# The Distortion of Binomial Voting Defies Expectation[*]

**Yannai A. Gonczarowski**[†]     **Gregory Kehne**[‡]     **Ariel D. Procaccia**[§]

**Ben Schiffer**[¶]          **Shirley Zhang**[‖]

## Abstract

In computational social choice, the *distortion* of a voting rule quantifies the degree to which the rule overcomes limited preference information to select a socially desirable outcome. This concept has been investigated extensively, but only through a worst-case lens. Instead, we study the *expected distortion* of voting rules with respect to an underlying distribution over voter utilities. Our main contribution is the design and analysis of a novel and intuitive rule, *binomial voting*, which provides strong distribution-independent guarantees for both expected distortion and expected welfare.

## 1 Introduction

In an election, voters report their preferences by casting ballots. Under the ubiquitous plurality rule, each voter names a single alternative, whereas other rules — such as the badly named *ranked-choice voting*,[7] whose adoption is rapidly expanding in the United States — require voters to rank the alternatives. However, even these ostensibly expressive ordinal ballots (whereby voters rank the alternatives) fail to capture voters' *intensity* of preference under truthful reporting.

If voters could report utility functions that are comparable to each other, then we would want to select socially desirable alternatives with respect to these utilities, but it is typically impractical to expect voters to compute and report such utilities. This creates a tension between the limited information available to the voting rule (through the report of ordinal ballots only) and its goal (good outcomes with respect to latent cardinal utilities).

A significant body of work in computational social choice aims to understand and alleviate this tension [Anshelevich et al., 2021]. It revolves around the notion of *distortion*, defined as the worst-case ratio between the utilitarian *social welfare* (sum of utilities) of the voting rule's outcome and that of the welfare-maximizing alternative. The worst case is taken over rankings, which serve as input to the voting rule, and over utilities that are consistent with these rankings.

---

[*]Gonczarowski gratefully acknowledges research support by the Harvard FAS Inequality in America Initiative. Procaccia gratefully acknowledges research support by the National Science Foundation under grants IIS-2147187, IIS-2229881 and CCF-2007080; and by the Office of Naval Research under grant N00014-20-1-2488. Schiffer was supported by an NSF Graduate Research Fellowship. Zhang was supported by an NSF Graduate Research Fellowship.

[†]Department of Economics and Paulson School of Engineering and Applied Sciences, Harvard University | *E-mail*: yannai@gonch.name.

[‡]Department of Computer Science, University of Texas at Austin | *E-mail*: gkehne@utexas.edu.

[§]Paulson School of Engineering and Applied Sciences, Harvard University | *E-mail*: ariel-pro@seas.harvard.edu.

[¶]Department of Statistics, Harvard University | *E-mail*: bschiffer1@g.harvard.edu.

[‖]Paulson School of Engineering and Applied Sciences, Harvard University | *E-mail*: szhang2@g.harvard.edu.

[7]Also known as "instant-runoff voting" or "alternative vote."

As is often the case with worst-case analysis, however, the classic notion of distortion is arguably too conservative. In particular, nontrivial guarantees require restrictive assumptions (see Section 1.2), and so this type of analysis may not help identify appealing voting rules.

With this difficulty in mind, we focus on *expected distortion*. Its definition includes the same ratio as before, and we are still interested in the worst case over rankings. However, we now take the conditional expectation over utilities consistent with the rankings, given an i.i.d. distribution over the utilities. Of course, it might not be realistic for such a distribution to be known a priori,[8] and thus we search for voting rules that are *distribution independent*[9]: rules that choose the outcome in a way that does not depend on this distribution, even though their guarantees are stated in terms of this distribution.

Overall, our goal is to **design distribution-independent voting rules that provide appealing expected distortion guarantees.** We uncover novel, potentially practical, voting rules with these properties.

## 1.1 Our Results

We start by considering the important case of two alternatives (e.g., US presidential elections or yes/no decisions) in Section 3. We show that, for any underlying distribution, the *majority* rule optimizes both expected distortion and expected social welfare.

This result suggests that maximizing expected social welfare may be a good approach for optimizing expected distortion. Indeed, our main result in Section 4 is that, under mild conditions on the underlying distribution and for a sufficiently large number of voters *or* alternatives, the expected-welfare-maximizing rule optimizes expected distortion almost perfectly.

The expected-welfare-maximizing rule, however, is tailored to the underlying distribution and relies on intimate knowledge thereof. Our aim is therefore to approximately optimize expected welfare via a distribution-independent rule.

In Section 5, we design and analyze such a rule, which belongs to the family of (positional) *scoring rules*. Under this rule, which we call *binomial voting*, each voter awards $\sum_{\ell=k}^{m} \binom{m}{\ell}$ points to the alternative ranked in the $k$th position, where $m$ is the number of alternatives, and the alternative awarded the largest number of points overall is selected; note that this rule is distribution independent. Our main result of Section 5 is that for any underlying distribution supported on $[0, 1]$ with (largest) median $\nu$, binomial voting provides a multiplicative $\frac{\nu}{2}$-approximation to the optimal expected welfare. Combining this result with that of Section 4, it follows that binomial voting gives almost the same $\frac{\nu}{2}$ guarantee for expected distortion, when the number of voters or alternatives is sufficiently large.

It is worth noting that binomial voting is *not* an outlandish rule designed purely to achieve low expected distortion. On the contrary: as a positional voting rule, it inherits the desirable properties of this family. In fact, positional scoring rules are characterized by a number of natural axioms [Young, 1975]. Furthermore, we are aware of very few positional scoring rules that have received attention in their own right, as it is typically difficult to justify any specific choice of scores; in this sense, the expected distortion framework can be seen as a way of pinpointing particularly useful parameters. In summary, we identify binomial voting as an unusually attractive rule when viewed through the lens of expected distortion.

## 1.2 Related Work

The literature on (worst-case) distortion [Anshelevich et al., 2021] can generally be partitioned into two threads. In the first thread [Procaccia and Rosenschein, 2006, Boutilier et al., 2015, Caragiannis et al., 2017, Mandal et al., 2019, Ebadian et al., 2022], it is assumed that voters have normalized utilities, that is, for each voter, the sum of utilities is one. Even with this restrictive assumption, deterministic voting rules cannot give nontrivial distortion bounds [Caragiannis et al., 2017], and the best possible distortion for randomized rules is $\Theta(1/\sqrt{m})$ [Ebadian et al., 2022]. In the second thread [Anshelevich and Postl, 2017, Gross et al., 2017, Anshelevich et al., 2018, Gkatzelis et al., 2020, Kizilkaya and Kempe, 2022], known as *metric distortion*, it is assumed that utilities (rather, costs) are induced by an underlying metric space. While some well-known voting rules have constant

---

[8]For this reason, we avoid calling this distribution a Bayesian *prior*.

[9]This term is inspired by the literature on prior-independent mechanisms within mechanism design.

distortion in this setting [Anshelevich et al., 2018], the metric assumption is arguably difficult to justify in most domains of interest. By contrast, we make no assumptions on utilities.

Our work is most closely related to that of Boutilier et al. [2015]. While their most substantial results pertain to worst-case distortion, one of their results deals with a distributional setting that can be seen as the starting point for our work. They show that the voting rule that maximizes expected social welfare is a scoring rule whose scores depend on the underlying distribution; we will revisit and build on this result. However, they do not study expected distortion, nor do they explore (in this context) voting rules that are agnostic to the distribution.

Previous papers that analyze expected distortion include those of Cheng et al. [2017, 2018]. However, their papers are fundamentally different. For one, they study metric distortion. More importantly, they focus on one intuitive but very specific distribution, where the positions of alternatives in the underlying metric space are drawn uniformly at random from the voter positions. By contrast, we study general (i.i.d.) distributions over utilities and design distribution-independent voting rules. The work of Ghodsi et al. [2019] is more distantly related: they also analyze expected distortion in the metric setting, assuming that voters abstain with some probability.

By replacing worst-case analysis with expectation, our work introduces a Bayesian view canonical to economic theory into the literature on distortion. The Wilson doctrine [Wilson, 1987] advocates for the use of mechanisms that require as little prior information as possible. Distribution-independent (sometimes called prior-independent) mechanisms are, through this lens, the most desirable ones as they require no prior information at all, and have been studied within economics [see, e.g., McAfee, 1992, Segal, 2003] as well as within computer science [see, e.g., Hartline and Roughgarden, 2009, Devanur et al., 2011, Babaioff et al., 2018]. Our work can also be seen as belonging to a recent push within computer science on "beyond worst-case" analysis of algorithms [Roughgarden, 2021].

## 2 Model and Preliminaries

Let there be a set $N = \{1, 2, \ldots, n\}$ of *voters* and a set $A = \{1, 2, \ldots, m\}$ of *alternatives*. Each voter $i$ has *utility* $u_{ij}$ for each alternative $j$. The voter utilities collectively form a *utility profile* $\mathbf{u}$, and the utilitarian *social welfare* for alternative $j$ is denoted as $\text{sw}(j, \mathbf{u}) = \sum_{i \in N} u_{ij}$. Each voter's utilities induce a *ranking* $\sigma_i = \sigma_i(\mathbf{u})$ over alternatives, where $j$ is ranked higher than $k$ in $\sigma_i$ only if $u_{ij} \geq u_{ik}$.[10] Let $\boldsymbol{\sigma} = (\sigma_1, \ldots, \sigma_n)$ be the *preference profile* consisting of the rankings of all voters.

A (deterministic) *voting rule* $f$ takes as input a preference profile and outputs a winning alternative $f(\boldsymbol{\sigma}) \in A$. One can also define a randomized voting rule, which returns a distribution over alternatives. The definitions given below easily generalize to randomized rules, by considering their expected social welfare, where the expectation is taken over the randomness of the rule. We focus on deterministic rules, however, for two reasons: first, for ease of exposition, and second, because deterministic rules are in fact as powerful as randomized rules in our setting, as we show in Appendix A.

The *worst-case distortion* of $f$ intuitively measures the loss in social welfare incurred by $f$, and is given by:

$$\sup_{\mathbf{u}} \frac{\max_{j \in A} \text{sw}(j, \mathbf{u})}{\text{sw}(f(\boldsymbol{\sigma}(\mathbf{u})), \mathbf{u})}. \tag{1}$$

The numerator is the optimal social welfare, whereas the denominator is the social welfare of the outcome under $f$; the worst case of this ratio is taken over utility profiles $u$. Note that a voting rule $f$ minimizes this expression if and only if it maximizes its inverse, and, more generally, that inverting this ratio does not change any of the results for worst-case distortion.

### 2.1 Distributional and Expected Distortion

Our point of departure is that we introduce an "average-case" setting; for the rest of this paper, we will assume that the voters' utilities for the alternatives are drawn i.i.d. from an underlying distribution $\mathcal{D}$. Unless otherwise noted, we will also assume that $\mathcal{D}$ is supported on $[0, 1]$. We will frequently use $\mu$ and $s^2$ to denote the mean and variance of $\mathcal{D}$, respectively.

---

[10]Tie breaking can be performed arbitrarily, and is of no consequence for our results.

Let $\mathbf{u} \triangleright \boldsymbol{\sigma}$ denote that the utility vector $\mathbf{u}$ is consistent with the preference profile $\boldsymbol{\sigma}$. Given a preference profile $\boldsymbol{\sigma}$, a utility vector $\mathbf{u} \triangleright \boldsymbol{\sigma}$ is drawn with respect to the distribution $\mathcal{D}$ as follows. Each voter $i \in N$ draws $m$ i.i.d. utilities from $\mathcal{D}$, and these $m$ utilities are assigned from highest to lowest to the $m$ alternatives according to the order of $\boldsymbol{\sigma}_i$, the ranking of voter $i$ under $\boldsymbol{\sigma}$. We define the *distributional distortion* under $\mathcal{D}$ of a voting rule $f$ for a preference profile $\boldsymbol{\sigma}$ as the following random variable, in the probability space defined by the above construction (for drawing $\mathbf{u} \triangleright \boldsymbol{\sigma}$ with respect to the distribution $\mathcal{D}$):

$$\mathrm{ddist}(f, \boldsymbol{\sigma}) = \frac{\mathrm{sw}(f(\boldsymbol{\sigma}), \mathbf{u})}{\max_{j \in A} \mathrm{sw}(j, \mathbf{u})}. \tag{2}$$

Note that this random variable representation of distributional distortion is in contrast to the single number representation in worst-case distortion. For the rest of this paper, when we refer to the *distortion* of a rule $f$, we are referring to the distributional distortion. We define the *expected distortion* of $f$ for $\boldsymbol{\sigma}$ as $\mathbb{E}\big[\mathrm{ddist}(f, \boldsymbol{\sigma})\big]$.

In Equation (2), we have defined distortion with the maximum in the denominator of the fraction. This means that the expected distortion of a voting rule lies in the range $[0, 1]$, and a larger value of expected distortion (closer to $1$) corresponds to a better voting rule. Note that this choice stands in contrast to Equation (1). As noted previously, inverting the ratio does not change the optimal rule $f$ for worst-case distortion. However, this no longer holds true when considering expected distortion. For a random variable $X$, in general $\mathbb{E}\big[\frac{1}{X}\big] \neq \frac{1}{\mathbb{E}[X]}$, and therefore rules that minimize $\mathbb{E}[X]$ can look very different compared to rules that maximize $\mathbb{E}\big[\frac{1}{X}\big]$. The reason for our choice is twofold. First, if the ratio were inverted then the random variable would be unbounded, which would imply that its expectation could be infinite even when the distribution $\mathcal{D}$ is bounded on $[0, 1]$. It turns out that this might be the case even for nice distributions such as beta distributions, as we prove in Appendix B. Second, if the ratio were inverted then there would exist distributions for which the best deterministic rule performs arbitrarily worse than the best randomized rule, even in situations where we would normatively expect the two rules to behave the same. That would prevent us from focusing on deterministic rules, as we are able to do here. For more details, see Appendix C.

We define an *expected-distortion-maximizing rule* (henceforth, an EDMR) as a rule $f$ that, for every preference profile $\boldsymbol{\sigma}$, maximizes the expected distortion under that profile. In other words, an expected-distortion-maximizing rule $f$ satisfies

$$\mathbb{E}\big[\mathrm{ddist}(f, \boldsymbol{\sigma})\big] = \max_g \mathbb{E}\big[\mathrm{ddist}(g, \boldsymbol{\sigma})\big]$$

for all preference profiles $\boldsymbol{\sigma}$, where the maximum is taken over all possible voting rules $g$. We write $\mathrm{edmr}(\cdot)$ to refer to an EDMR.

## 2.2 Distributional and Expected Social Welfare

Similarly to Boutilier et al. [2015], we also wish to consider expected social welfare. To this end, we define the *distributional social welfare* of a rule $f$ for preference profile $\boldsymbol{\sigma}$ as the random variable

$$\mathrm{dsw}(f, \boldsymbol{\sigma}) = \mathrm{sw}(f(\boldsymbol{\sigma}), \mathbf{u}),$$

in the same probability space as before (defined by the construction for drawing $\mathbf{u} \triangleright \boldsymbol{\sigma}$ with respect to a distribution $\mathcal{D}$). We define the *expected social welfare* of $f$ for $\boldsymbol{\sigma}$ as $\mathbb{E}\big[\mathrm{dsw}(f, \boldsymbol{\sigma})\big]$.

An *expected-welfare-maximizing rule* (henceforth, an EWMR) is a rule $f$ that, for all preference profiles $\boldsymbol{\sigma}$, maximizes the expected social welfare, i.e., $f$ such that for all $\boldsymbol{\sigma}$,

$$\mathbb{E}\big[\mathrm{dsw}(f, \boldsymbol{\sigma})\big] = \max_g \mathbb{E}\big[\mathrm{dsw}(g, \boldsymbol{\sigma})\big].$$

We write $\mathrm{ewmr}(\cdot)$ to refer to an EWMR.

The family of (positional) *scoring rules* consists of voting rules that assign a vector of scores $(s_1, ..., s_m)$ to alternatives, from highest ranked through lowest ranked, respectively. The alternative that is chosen is the alternative that has the highest total score when adding up all of the rankings across the $n$ voters, with ties broken arbitrarily. Boutilier et al. [2015] show that a scoring rule using scores

$$(s_1, ..., s_m) = \big(\mathbb{E}[X_{(m)}], \mathbb{E}[X_{(m-1)}], ..., \mathbb{E}[X_{(1)}]\big)$$

is an EWMR, where $\mathbb{E}\big[X_{(m-k)}\big]$ is the expected value of the $m-k$ order statistic from $m$ i.i.d. draws from $\mathcal{D}$. Note, however, that this rule is tailored to the specific distribution $\mathcal{D}$.

## 2.3 Approximately Optimal Rules

The goal of this paper is to explore rules that are approximately as good as an EDMR, but are distribution independent, i.e., are not tailored to $\mathcal{D}$. This will rely on approximations to an EWMR. We call a voting rule $f$ an $\alpha$-*Expected-Distortion-Maximizing Rule* ($\alpha$-EDMR) for distribution $\mathcal{D}$ if, for all preference profiles $\boldsymbol{\sigma}$, the expected distortion of $f$ is at least $\alpha$ times the expected distortion of an EDMR. In other words, $f$ is an $\alpha$-EDMR for $\mathcal{D}$ if for all $\boldsymbol{\sigma}$,

$$\mathbb{E}\big[\mathrm{ddist}(f, \boldsymbol{\sigma})\big] \geq \alpha \cdot \mathbb{E}\big[\mathrm{ddist}(\mathrm{edmr}, \boldsymbol{\sigma})\big].$$

If a voting rule is a $(1 - \varepsilon)$-EDMR, then we refer to it as an $\varepsilon$-*approximation* of an EDMR.

We use the analogous terminology of $\alpha$-EWMR for rules $f$ such that, for all $\boldsymbol{\sigma}$,

$$\mathbb{E}\big[\mathrm{dsw}(f, \boldsymbol{\sigma})\big] \geq \alpha \cdot \mathbb{E}\big[\mathrm{dsw}(\mathrm{ewmr}, \boldsymbol{\sigma})\big].$$

To make our goal more tangible, consider a policy maker in charge of choosing a voting rule for a certain municipality. There are various guarantees such a policy maker might wish for the voting rule to satisfy with respect to (expected) distortion. If said policy maker is choosing a voting rule to use for years to come, they are likely to be interested in distribution-independent guarantees, since demographic distributions might shift over time.

A first guarantee in which such a policy maker might be interested is that *ex ante*, before an election is held, the distortion is expected to be low. Note that this expectation is over the entire space of voter utility profiles. While this is an appealing guarantee, it is rather weak (and easy to satisfy). To see the sense in which it is weak, consider the policy maker not at the day on which they choose the voting rule, but rather a while later, in a specific election that uses this rule, after the votes have been cast and officially tallied. At this point in time, the ordinal preferences $\boldsymbol{\sigma}$ are public knowledge, and, depending on what they are, it might be the case that despite the expected distortion having been low *ex ante*, it turns out that the expected distortion is high *ex post*, that is, it turns out that conditioned on all of the public information so far—i.e., conditioned on the *ordinal* preference profile $\boldsymbol{\sigma}$—the (conditioned) expected distortion is high. While the policy maker had good reasons to choose the voting rule *ex ante*, a news outlet might still run a news story noting that given the tallied (realized, possibly low probability) ordinal preferences $\boldsymbol{\sigma}$, there is reason to expect very high distortion. A forward-looking (and negative-press-averse) policy maker might want a (worst-case) guarantee that this scenario cannot happen, i.e., a worst-case guarantee on the *ex post* expected distortion. This is of course a much stronger guarantee (in particular, it also implies the same guarantee on *ex ante* expected distortion), and is considerably harder to satisfy. Seeking rules that give such a guarantee is precisely the goal of this paper.

We note that such an *ex post* guarantee is in a sense the strongest guarantee that such a negative-press-averse policy maker might wish for. Indeed, an even stronger guarantee would be for the distortion to be low given the actual realized *cardinal* preferences $\mathbf{u}$, which coincides with "traditional" worst-case distortion, but since the policy maker cares about press coverage, and since the realized cardinal preferences never become public knowledge (and hence, never become the basis for a news story), such a strong guarantee would be an overkill for the policy maker. This is of course fortunate, because as noted in Section 1.2, there are impossibilities that preclude getting such an overly strong guarantee in a distribution-independent setting like ours.

## 3 The Case of Two Alternatives

To develop some intuition, let us start with a naïve attempt to compute an EDMR: calculate the expected distortion for each alternative $j \in A$, and choose the alternative that optimizes the expected distortion. This algorithm has two major shortcomings, however. First, calculating the expected distortion of a fixed alternative requires $nm$ concentric integrals, and therefore this algorithm would take time that is exponential in both $n$ and $m$. Second, the naïve algorithm requires knowledge of the underlying distribution $\mathcal{D}$.

Even for the case of two alternatives, the above algorithm is exponential in $n$ and therefore not computationally feasible—not to mention that it still requires knowledge of the distribution. However, it turns out that for this special case, *majority*—which selects the alternative preferred by the majority of voters (with ties broken arbitrarily)—is both an EDMR and an EWMR. Majority is furthermore computationally efficient and distribution-independent (i.e., agnostic to the distribution), so it satisfies all of the desiderata discussed so far. As a bonus, it is also indisputably practical.

**Theorem 3.1** *Let $m = 2$. For every distribution $\mathcal{D}$, majority is both an expected-distortion-maximizing rule and an expected-welfare-maximizing rule.*

**Proof.** We first show that majority is an EWMR. Let $\mu_1$ and $\mu_2$ be the expected values for the minimum and maximum, respectively, of two i.i.d. draws from $\mathcal{D}$. Let $0 \leq k \leq n$ be the number of voters that prefer alternative 1 to alternative 2. Then, by linearity of expectation, the expected welfares for alternative 1 and alternative 2 are $k\mu_2 + (n-k)\mu_1$ and $(n-k)\mu_2 + k\mu_1$, respectively. Since $\mu_2 \geq \mu_1$, alternative 1 is an expected-welfare-maximizing alternative if and only if $k \geq n - k$.

To prove that majority is an EDMR, we use a coupling argument to show that the expected distortion of an alternative that is ranked first by $k$ out of $n$ voters is increasing in $k$. A direct consequence of this is that a majority winner is an expected-distortion-maximizing alternative.

Recall that $u_{ij}$ is defined as the utility that voter $i$ has for alternative $j$ for $i \in N$ and $j \in \{1, 2\}$. Let $\boldsymbol{\sigma}^k$ be the preference profile for $n$ voters that has alternative 1 ranked first by the first $k$ voters (and only by them). Similarly, let $\boldsymbol{\sigma}^{k+1}$ be the preference profile for $n$ voters that has alternative 1 ranked first by the first $k$ voters and by the last voter (and only by them). To show that the majority winner is an EDMR, it is sufficient (by symmetry) to show that

$$
\mathop{\mathbb{E}}_{\mathbf{u} \triangleright \boldsymbol{\sigma}^k} \left[ \frac{\sum_{i=1}^n u_{i1}}{\max\{\sum_{i=1}^n u_{i1}, \sum_{i=1}^n u_{i2}\}} \right] \leq \mathop{\mathbb{E}}_{\mathbf{u} \triangleright \boldsymbol{\sigma}^{k+1}} \left[ \frac{\sum_{i=1}^n u_{i1}}{\max\{\sum_{i=1}^n u_{i1}, \sum_{i=1}^n u_{i2}\}} \right].
$$

Note that we used $\mathbb{E}_{\mathbf{u} \triangleright \boldsymbol{\sigma}^k}$ to denote the expectation over a draw of a utility profile $\mathbf{u}$ consistent with $\boldsymbol{\sigma}^k$, and similarly for $k+1$.

Next, define $Z_j = \sum_{i=1}^{n-1} u_{ij}$ for $j \in \{1, 2\}$. Furthermore, define $\boldsymbol{\sigma}^k_{-n}$ as a preference profile on $n-1$ voters that has the first $k$ voters ranking alternative 1 higher than alternative 2. Finally, let $\mathbf{u}_{-n}$ be the truncated version of $u$ that excludes the $n$th voter. By the law of total expectation, we can take an outer expectation over the utilities of the first $n-1$ voters and an inner expectation over the utilities of the $n$th voter:

$$
\mathop{\mathbb{E}}_{\mathbf{u} \triangleright \boldsymbol{\sigma}^k} \left[ \frac{\sum_{i=1}^n u_{i1}}{\max\{\sum_{i=1}^n u_{i1}, \sum_{i=1}^n u_{i2}\}} \right] = \mathop{\mathbb{E}}_{\mathbf{u}_{-n} \triangleright \boldsymbol{\sigma}^k_{-n}} \left[ \mathop{\mathbb{E}}_{u_{n1} \leq u_{n2}} \left[ \frac{Z_1 + u_{n1}}{\max\{Z_1 + u_{n1}, Z_2 + u_{n2}\}} \,\middle|\, \mathbf{u}_{-n} \right] \right],
$$

$$
\mathop{\mathbb{E}}_{\mathbf{u} \triangleright \boldsymbol{\sigma}^{k+1}} \left[ \frac{\sum_{i=1}^n u_{i1}}{\max\{\sum_{i=1}^n u_{i1}, \sum_{i=1}^n u_{i2}\}} \right] = \mathop{\mathbb{E}}_{\mathbf{u}_{-n} \triangleright \boldsymbol{\sigma}^k_{-n}} \left[ \mathop{\mathbb{E}}_{u_{n1} \geq u_{n2}} \left[ \frac{Z_1 + u_{n1}}{\max\{Z_1 + u_{n1}, Z_2 + u_{n2}\}} \,\middle|\, \mathbf{u}_{-n} \right] \right].
$$

By the definition of $\boldsymbol{\sigma}^k$ and $\boldsymbol{\sigma}^{k+1}$, the two final expressions only differ in the inequality between the random variables $u_{n1}$ and $u_{n2}$. Furthermore, since $Z_1$ and $Z_2$ are both functions of $\mathbf{u}_{-n}$, they are both constants in the inner expectation. Therefore, to prove the desired result it is sufficient to show that for any constants $Z_1, Z_2 \geq 0$,

$$
\mathop{\mathbb{E}}_{u_{n1} \leq u_{n2}} \left[ \frac{Z_1 + u_{n1}}{\max\{Z_1 + u_{n1}, Z_2 + u_{n2}\}} \right] \leq \mathop{\mathbb{E}}_{u_{n1} \geq u_{n2}} \left[ \frac{Z_1 + u_{n1}}{\max\{Z_1 + u_{n1}, Z_2 + u_{n2}\}} \right].
$$

Assume first that $f$ is a continuous distribution. Let $f$ be the PDF of $\mathcal{D}$. The joint PDF of the minimum and maximum of two draws from $\mathcal{D}$ is $2f(x)f(y)$ for $x \geq y$. This means that the expectation can be rewritten as a double integral giving the desired result:

$$
\begin{aligned}
\mathop{\mathbb{E}}_{u_{n1} \geq u_{n2}} \left[ \frac{Z_1 + u_{n1}}{\max\{Z_1 + u_{n1}, Z_2 + u_{n2}\}} \right] &= \int_0^\infty \int_y^\infty \frac{Z_1 + x}{\max\{Z_1 + x, Z_2 + y\}} 2f(x)f(y)dxdy \\
&\geq \int_0^\infty \int_y^\infty \frac{Z_1 + y}{\max\{Z_1 + y, Z_2 + x\}} 2f(x)f(y)dxdy \\
&= \mathop{\mathbb{E}}_{u_{n1} \leq u_{n2}} \left[ \frac{Z_1 + u_{n1}}{\max\{Z_1 + u_{n1}, Z_2 + u_{n2}\}} \right].
\end{aligned}
$$

The inequality in the second line follows from a simple lemma in Appendix D, which takes advantage of the fact that $x \geq y$.

A similar derivation using the Lebesgue decomposition or the Riemann–Stieltjes integral gives the same result for general (not necessarily continuous) $f$. ∎

When $m \geq 2$, majority is no longer defined, and simple extensions (like plurality) are neither an EDMR nor an EWMR. We therefore require a different strategy for such $m$.

## 4 From Expected Distortion to Expected Welfare

Despite failing to extend beyond the case of two alternatives, Theorem 3.1 does give some hope that perhaps an expected-welfare-maximizing rule, which is equivalent to an expected-distortion-maximizing rule when $m = 2$, can provide good expected distortion for larger $m$. This is not the case in general, however: If we consider the space of all distributions $\mathcal{D}$ supported on $[0, 1]$, an EWMR may not provide a good approximation to an EDMR. In fact, there exist distributions and preference profiles for which the expected distortion of an EWMR is an $\Omega(m)$ factor worse than that of an EDMR, as shown in Appendix E.

**Theorem 4.1** *There exists a constant $C > 0$ such that for every $m \geq 5$ there exist a number of voters $n$, a distribution $\mathcal{D}$ supported on $[0, 1]$, and a preference profile $\boldsymbol{\sigma}$, such that $\mathbb{E}\big[\mathrm{ddist}(\mathrm{ewmr}, \boldsymbol{\sigma})\big] \leq \frac{C}{m} \cdot \mathbb{E}\big[\mathrm{ddist}(\mathrm{edmr}, \boldsymbol{\sigma})\big]$.*

It is worth noting that the rule that chooses one of the $m$ alternatives uniformly at random has expected distortion no more than a factor $O(m)$ wose than that of an EDMR. Therefore, this theorem shows that for some distributions and preference profiles, an EWMR performs almost as poorly as the rule that chooses an alternative uniformly at random.

One distinctive trait of the distribution used to prove Theorem 4.1 is that both the values that the distribution can take and their respective probabilities depend on $n$ and $m$. This is perhaps unnatural, as we might expect that as the numbers of voters and alternatives increase, the i.i.d. distribution from which each voter draws utilities is fixed. In fact, with such an assumption that the distribution $\mathcal{D}$ is independent of $n$ and $m$, an EWMR will be an $\varepsilon$-approximation for an EDMR for sufficiently large $n$ (regardless of the value of $m$). Furthermore, with the additional assumptions that the distribution is continuous and that the derivative of the CDF is bounded from below, an EWMR will also be an $\varepsilon$-approximation for an EDMR for sufficiently large $m$ (regardless of the value of $n$). Therefore, asymptotically in both $n$ and $m$, an EWMR does perform well by the metric of expected distortion; the proof is relegated to Appendix F.

**Theorem 4.2** *Assume $\mathcal{D}$ is supported on $[0, 1]$ and has constant (relative to $n, m$) mean $\mu$ and variance $s^2$. Then for every $\varepsilon > 0$, there exists $n_0$ such that if $n \geq n_0$, then $\mathbb{E}\big[\mathrm{ddist}(\mathrm{ewmr}, \boldsymbol{\sigma})\big] \geq (1 - \varepsilon)\mathbb{E}\big[\mathrm{ddist}(\mathrm{edmr}, \boldsymbol{\sigma})\big]$.*

*Furthermore, if $\mathcal{D}$ has a continuously differentiable CDF $F$ satisfying $\frac{dF}{dx} > 0$, then there exists $m_0$ such that the same result holds if $m \geq m_0$.*

Note that it is not a priori obvious that either large $n$ or large $m$ are sufficient conditions in Theorem 4.2, and in fact the proofs rely on different forms of concentration. For large $n$, the key to the proof is that sums of i.i.d. bounded random variables concentrate by Hoeffding's inequality. For large $m$, the key idea is that, for continuous distributions, the variance of the order statistics goes to 0 as the number of samples $m$ grows.

## 5 Distribution-Independent Expected-Welfare Maximization

Theorem 4.2 gives a computationally simple way to calculate an "almost EDMR" for fixed $\mu$ and $s^2$ by simply calculating an EWMR, which—as noted in Section 2—can be done by using a scoring rule based on the expected order statistics of $\mathcal{D}$. This is satisfying because it has a practical form and performs well for both metrics, namely expected distortion and expected welfare. A fundamental flaw with such a rule, however, is that it relies on knowing the entire underlying distribution $\mathcal{D}$, which might not be realistic in practice. Our strategy for approximating an EDMR, therefore, is to seek a *distribution-independent* approximation to an EWMR, that is, a rule that simultaneously provides

guarantees for every distribution (at least under some assumptions). To this end, we leverage the following observation.

**Observation 5.1** *If a voting rule $f$ is a $\beta$-EWMR, and an EWMR is a $\gamma$-EDMR, then $f$ is a $\beta\gamma$-EDMR.*

In particular, for any $\varepsilon > 0$, Theorem 4.2 yields $\gamma = (1 - \varepsilon)$ for sufficiently large $n$ or $m$. Our goal in this section, therefore, is to seek a rule $f$ for which we can establish a value of $\beta$ in a distribution-independent way.

Our first result is negative: It is impossible for a single voting rule to achieve high expected social welfare for all distributions supported on $[0, 1]$.

**Theorem 5.2** *For every $0 < \alpha \leq 1$ there exist $n, m, \boldsymbol{\sigma}$ and two distributions $\mathcal{D}_1, \mathcal{D}_2$ that are both supported on $[0, 1]$ such that no alternative is an $\alpha$-EWMR for both $D_1$ and $D_2$ under $\boldsymbol{\sigma}$.*

We prove Theorem 5.2 in Appendix G. We will circumvent Theorem 5.2 in two natural ways. Our first way will do so by restricting the class of distributions. Specifically, we first focus on the class of *symmetric* distributions supported on $[0, 1]$, that is, distributions $\mathcal{D}$ such that $\Pr_{x \sim \mathcal{D}}\left(x \leq \frac{1}{2} - \varepsilon\right) = \Pr_{x \sim \mathcal{D}}\left(x \geq \frac{1}{2} + \varepsilon\right)$ for all $\varepsilon \in \left[0, \frac{1}{2}\right]$.

The fact that an EWMR is given by a scoring rule [Boutilier et al., 2015], albeit one that depends on the distribution, suggests that defining a scoring rule in a distribution-independent way may be a good approach. Consider the following scoring rule, whereby voters give their approval to the top half of their ranking.

**Definition 5.3** Top-half approval *is the scoring rule that assigns a score of $1$ to alternatives ranked in the highest $\left\lceil \frac{m}{2} \right\rceil$ positions and $0$ to all other alternatives.*

It turns out that top-half approval gives good guarantees for the subclass of symmetric distributions, as we prove in Appendix H.

**Theorem 5.4** *For all $n, m$, top-half approval is a $\frac{1}{3}$-EWMR for all symmetric distributions $\mathcal{D}$.*

Given the attractiveness of top-half approval, a natural follow-up question is whether another distribution-independent rule can give an even better EWMR approximation for every symmetric distribution. While we cannot preclude this possibility altogether, we do give an upper bound (negative result) on the feasible approximation in Appendix I, which holds not just for scoring rules but for any voting rule.

**Theorem 5.5** *No distribution-independent voting rule is an $\alpha$-EWMR for all symmetric distributions $\mathcal{D}$, for any $\alpha > \sqrt{\frac{1}{3}}$.*

## 5.1 Binomial Voting

Top-half approval clearly leaves some information on the table, as it ignores the ranking of alternatives within each half. Let us refine it, then, by considering a variant that accounts for this additional information. As we will see, this refinement will allow us to dispense with the assumption of a symmetric distribution.

**Definition 5.6** Binomial voting *is the scoring rule that assigns a score of $\sum_{\ell=k}^{m} \binom{m}{\ell}$ to an alternative that is ranked $k$.*

Binomial voting can be seen as a "smoother" version of top-half approval. Indeed, because the binomial coefficients are largest in the vicinity of $\frac{m}{2}$, the top, say, $\frac{m}{2} - \sqrt{m}$ positions have scores close to the maximum, and the bottom $\frac{m}{2} - \sqrt{m}$ positions have scores close to the minimum, with the transition occurring rapidly in between.

Our main result for binomial voting is the following, which we prove in in Appendix J. This result circumvents Theorem 5.2 not by restricting the class of distributions, but rather by giving a guarantee that depends on the median of the distribution.

**Theorem 5.7** *Let $\mathcal{D}$ be a distribution supported on $[0,1]$ whose largest median is $\nu$, i.e., $\nu = \sup\{y \mid \Pr_{x \sim \mathcal{D}}(x \leq y) \leq \frac{1}{2}\}$. Then binomial voting is a $\frac{\nu}{2}$-EWMR for $\mathcal{D}$.*

Note that any symmetric distribution has a largest median of $\nu \geq \frac{1}{2}$, so for such distributions Theorem 5.7 almost recovers the guarantee given by top-half approval for symmetric distributions. Needless to say, the guarantee for binomial voting (Theorem 5.7) is much more powerful than that for top-half approval (Theorem 5.4), as the former applies to any distribution and not only to symmetric ones. Note that while Theorem 5.7 gives a guarantee that depends on the median of the distribution $\mathcal{D}$, because binomial voting is distribution-independent, it achieves this guarantee even when nothing is known about the distribution, including its median.

By putting together Theorem 4.2, Observation 5.1, and Theorem 5.4, we get the following corollary, which ties these results back to expected distortion.

**Corollary 5.8** *Let $\mathcal{D}$ be supported on $[0,1]$ with constant (relative to $n, m$) mean, variance, and largest median $\nu$. Then for every $\varepsilon$, there exists $n_0$ such that if $n \geq n_0$, binomial voting is a $\left(\frac{1}{2} - \varepsilon\right)\nu$-EDMR for $\mathcal{D}$.*

*Furthermore, if $\mathcal{D}$ has a continuously differentiable CDF $F$ satisfying $\inf_x \frac{dF(x)}{dx} > 0$, then there exists $m_0$ such that the same result holds if $m \geq m_0$.*

### 5.2 Beyond the Median of the Distribution

A natural generalization of binomial voting is to consider quantiles other than the median. For example, one might wish to guarantee a $\beta$-EWMR where $\beta$ depends on the third quartile. The following corollary generalizes binomial voting to other distribution-free scoring rules with expected social welfare guarantees based on the quantile of interest. All proofs for this section can be found in Appendix K.

**Corollary 5.9** *Let $p \in (0,1]$. Consider the scoring rule that assigns a score of $\sum_{\ell=k}^{m} \binom{m}{\ell}(1-p)^\ell p^{m-\ell}$ to an alternative ranked $k$. This scoring rule is a $(1-p)Q$-EWMR, where $Q$ is the $p$-quantile of $\mathcal{D}$, i.e. $Q = \sup\{y \mid \Pr_{x \sim \mathcal{D}}(x \leq y) \leq p\}$.*

So far we have focused on distribution-independent rules where the rules stay the same but the bounds change depending on the underlying distribution. By contrast, the known EWMR of Boutilier et al. [2015] that requires complete knowledge about the underlying distributions looks very different for each distribution. It turns out that we can interpolate between these two extremes by incorporating partial information that might be known about the distribution $\mathcal{D}$. If we are given access to multiple quantiles of the distribution, then we can extend binomial voting to a distribution-dependent rule as follows.

**Definition 5.10** *Suppose we have access to quantiles $Q_1, \ldots, Q_T$ corresponding to fractions $0 < p_1 < p_2 < \cdots < p_T \leq 1$, where $Q_k = \sup\{y \mid \Pr_{x \sim \mathcal{D}}(x \leq y) \leq p_k\}$. Define generalized binomial voting as the scoring rule that gives to the alternative ranked $k$ a score of*

$$s_k := \sum_{t=1}^{T} Q_t \left( \sum_{\ell=k}^{m} \binom{m}{\ell}(1-p_t)^\ell p_t^{m-\ell} - \sum_{\ell=k}^{m} \binom{m}{\ell}(1-p_{t+1})^\ell p_{t+1}^{m-\ell} \right).$$

**Theorem 5.11** *If the distribution $\mathcal{D}$ has quantiles $Q_1, \ldots, Q_T$ corresponding to fractions $0 < p_1 < p_2 < \cdots < p_T \leq 1$ (define $p_{T+1} = 1$), then generalized binomial voting is a $\beta$-EWMR for $\beta = \sum_{s=1}^{T} Q_s(p_{s+1} - p_s)$.*

As an example, suppose the only information known about the underlying distribution is that the values $Q_1, Q_2, Q_3$ are the three quartiles (25%, 50%, 75%). Then generalized binomial voting is a $\frac{Q_1 + Q_2 + Q_3}{4}$-EWMR. Whenever $Q_3 > Q_2$ or $Q_1 > 0$, this is strictly stronger than vanilla binomial voting, which is guaranteed to be a $\frac{Q_2}{2}$-EWMR. Given more quantiles, generalized binomial voting gives weakly stronger approximations of an EWMR. Intuitively, generalized binomial voting is approximating the distribution $\mathcal{D}$ with left-oriented rectangles based on the quantile information. In fact, with sufficient quantile information, the rule will exactly approach an EWMR in the limit as $T \to \infty$. This result follows from generalized binomial voting being an approximation based on quantile rectangles, which in the limit becomes exactly the integral for expectation.

**Corollary 5.12** *For the quantiles $(p_1, p_2, ...p_T) = (\frac{1}{T}, \frac{2}{T}, ..., \frac{T}{T})$, generalized binomial voting approaches an expected-welfare-maximizing rule as $T \longrightarrow \infty$.*

# 6 Discussion

In this section we discuss limitations of our work and present some extensions thereof.

## 6.1 Limitations

In our view, the main limitation of our work is the simplifying assumption that voters' utilities for all alternatives are independent and identically distributed. In reality, some alternatives are inherently stronger than others. In addition, utilities are typically correlated; for example, in US politics, if a voter's favorite candidate is a Republican, they are likely to prefer another Republican candidate to a Democratic candidate.

That said, similar i.i.d. assumptions are commonly made in computational social choice and mechanism design [see, e.g., Dickerson et al., 2014]. Moreover, there is a large body of literature that relies on the *impartial culture assumption* [Tsetlin et al., 2003, Pritchard and Wilson, 2009], under which rankings are drawn independently and uniformly at random; when rankings are induced by utilities, the i.i.d. assumption on utilities gives rise to impartial culture. While these assumptions are admittedly strong, they lead to clean insights and solutions that often generalize well.

## 6.2 Benchmarks

We chose to define our benchmarks of expected distortion and expected welfare with respect to an arbitrary fixed preference profile $\sigma$. Our main results therefore hold for any choice of preference profile $\sigma$. In other words, our results should be interpreted as being in the worst-case regime for preference profiles while being in the average-case regime for utilities. We chose to use the EDMR and EWMR as benchmarks in our main theorems because, by definition, these are the rules which maximize our objectives. However, the proofs of our results do not fundamentally rely on the use of these benchmarks. In fact, many of our main results (including Theorem 4.2 and Theorem 5.7) can be adapted to directly lower bound the actual values of expected distortion and expected welfare. More details can be found in Appendix M.

## 6.3 Extensions

We have shown that an EWMR can be a good approximation to an EDMR. One follow-up question is whether it is possible to do better by directly computing an EDMR. As noted earlier, a potential EDMR is to choose the deterministic alternative with the highest expected distortion. Unfortunately, this rule requires exponential computation time. One way to circumvent this is to use i.i.d. samples from the underlying distribution to empirically approximate the expected distortion of each alternative, as shown in Appendix L. With polynomially many samples from $\mathcal{D}$ and polynomial time, such a rule is a $(1 - \varepsilon)$-EDMR with high probability. However, it may be unrealistic to expect i.i.d. sample access to the underlying distribution, which limits the practicality of this approach.

Theorem 5.2 states that for a given constant $\alpha$, no rule can be an $\alpha$-EWMR for all distributions. Despite this, one might hope that a rule that is an EWMR for one distribution could also be a good approximation for an EWMR for another similar distribution. This is in fact not the case. Specifically, for any $\alpha, \varepsilon > 0$, an EWMR for a distribution $\mathcal{D}$ may not even be an $\alpha$-EWMR for another distribution that has total variation distance only $\varepsilon$ away from $\mathcal{D}$. Surprisingly, though, a voting rule called Borda count, which is an EWMR for the uniform distribution, is in fact a $(1-\varepsilon)$-EWMR for all distributions that are sufficiently close to uniform. For more details, see Appendix N.

Finally, Theorem 5.7 gives a strong positive result for the expected welfare of binomial voting that depends on the median of the distribution. In Appendix O, we show that for any distribution (regardless of its median) and any $n, m$, plurality is an $\alpha$-EWMR for $\alpha = \max\{\frac{1}{n}, \frac{1}{m}\}$. We conjecture that plurality may be "tight" in the sense that no rule can provide a better guarantee, but we leave this question open for future work.

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

## A Randomized Rules

As mentioned in Section 2, the definition of distributional distortion (2) can be extended to randomized voting rules by defining it to be the random variable

$$\text{ddist}(f, \boldsymbol{\sigma}) := \frac{\mathbb{E}[\text{sw}(f(\boldsymbol{\sigma}), \mathbf{u})]}{\max_{j \in A} \text{sw}(j, \mathbf{u})}, \tag{3}$$

where the expectation in the numerator is taken only over randomness in the voting rule $f$. In this paper we choose to only consider the deterministic rules because linearity of expectation implies that any randomized voting rule $f$ is weakly worse than a deterministic rule. Therefore, although the number of randomized rules is infinite, considering the finite number of deterministic rules is sufficient for finding an EDMR. Note that the same argument holds for EWMR as well. This result is formalized in the following lemma.

**Lemma A.1** *For any randomized voting rule, there exists a deterministic rule that has expected distortion as least as large as that of the randomized rule. The same result holds for expected welfare.*

**Proof.** We will start by proving the result for expected distortion. The expected distortion of a randomized voting rule $f$ can be written as follows, where the inner expectation is over the randomness in the voting rule $f$ and the outer expectation is over the random utilities $\mathbf{u} \triangleright \boldsymbol{\sigma}$. Note that the randomness in $f$ is over the $m$ alternatives that $f$ can select.

$$\mathop{\mathbb{E}}_{\mathbf{u} \triangleright \boldsymbol{\sigma}} \left[ \frac{\mathbb{E}[\text{sw}(f(\boldsymbol{\sigma}), \mathbf{u})]}{\max_{k \in A} \text{sw}(k, \mathbf{u})} \right] = \mathop{\mathbb{E}}_{\mathbf{u} \triangleright \boldsymbol{\sigma}} \left[ \frac{\sum_{j=1}^{m} \Pr(f(\boldsymbol{\sigma}) = j) \cdot \text{sw}(j, \mathbf{u})}{\max_{k \in A} \text{sw}(k, \mathbf{u})} \right]$$

$$= \mathop{\mathbb{E}}_{\mathbf{u} \triangleright \boldsymbol{\sigma}} \left[ \sum_{j=1}^{m} \Pr(f(\boldsymbol{\sigma}) = j) \cdot \frac{\text{sw}(j, \mathbf{u})}{\max_{k \in A} \text{sw}(k, \mathbf{u})} \right]$$

$$= \sum_{j=1}^{m} \Pr(f(\boldsymbol{\sigma}) = j) \cdot \mathop{\mathbb{E}}_{\mathbf{u} \triangleright \boldsymbol{\sigma}} \left[ \frac{\text{sw}(j, \mathbf{u})}{\max_{k \in A} \text{sw}(k, \mathbf{u})} \right]$$

$$\leq \max_{j} \left( \mathop{\mathbb{E}}_{\mathbf{u} \triangleright \boldsymbol{\sigma}} \left[ \frac{\text{sw}(j, \mathbf{u})}{\max_{k \in A} \text{sw}(k, \mathbf{u})} \right] \right).$$

This completes the proof for expected distortion, as the final expression is a deterministic rule.

For expected welfare, the result follows in a similar manner:

$$\mathop{\mathbb{E}}_{\mathbf{u} \triangleright \boldsymbol{\sigma}} \left[ \mathbb{E}[\text{sw}(f(\boldsymbol{\sigma}), \mathbf{u})] \right] = \mathop{\mathbb{E}}_{\mathbf{u} \triangleright \boldsymbol{\sigma}} \left[ \sum_{j=1}^{m} \Pr(f(\boldsymbol{\sigma}) = j) \cdot \text{sw}(j, \mathbf{u}) \right]$$

$$= \sum_{j=1}^{m} \Pr(f(\boldsymbol{\sigma}) = j) \cdot \mathop{\mathbb{E}}_{\mathbf{u} \triangleright \boldsymbol{\sigma}} [\text{sw}(j, \mathbf{u})]$$

$$\leq \max_{j} \left( \mathop{\mathbb{E}}_{\mathbf{u} \triangleright \boldsymbol{\sigma}} [\text{sw}(j, \mathbf{u})] \right).$$

∎

## B Infinite Expected Inverse Distributional Distortion

As mentioned in Section 2, we chose to define distributional distortion with the ratio in the opposite direction as worst-case distortion. Another possible definition would have been to define the *inverse distributional distortion* as follows, which is exactly the inverse of distributional distortion. Note that we have defined inverse distributional distortion using the more general definition for randomized voting rules introduced above, as in (3).

**Definition B.1** *We define the* inverse distributional distortion *of a rule $f$ under preference profile $\boldsymbol{\sigma}$ as*

$$\mathrm{invddist}(f, \boldsymbol{\sigma}) = \frac{\max_{j \in A} \mathrm{sw}(j, \mathbf{u})}{\mathbb{E}[\mathrm{sw}(f(\boldsymbol{\sigma}), \mathbf{u})]}. \tag{4}$$

*where the expectation is over the the randomness of the voting rule.*

As before, we can consider the expectation of the random variable $\mathrm{invddist}(f, \boldsymbol{\sigma})$ with respect to a random $\mathbf{u} \triangleright \boldsymbol{\sigma}$. Unfortunately, even for relatively simple distributions supported on $[0, 1]$, the expectation of $\mathrm{invddist}(f, \boldsymbol{\sigma})$ can be infinite. This makes comparing rules more difficult when using expected inverse distortion as the metric.

**Lemma B.2** *Let $m = n = 2$ and $\boldsymbol{\sigma}$ be a symmetric preference profile. If $\mathcal{D} \sim \beta(\frac{1}{4}, 1)$, then no deterministic rule $f$ has finite $\mathbb{E}[\mathrm{invddist}(f, \boldsymbol{\sigma})]$.*

**Proof.** Let $f(x) = 0.25x^{-0.75}$ be the pdf of $\mathcal{D}$. Note that the joint pdf, denoted $f_{X_{(1)}X_{(2)}}(x, y)$, of the minimum $X_{(1)}$ and maximum $X_{(2)}$ of two i.i.d. draws from $\mathcal{D}$ is:

$$f_{X_{(1)}X_{(2)}}(x, y) = 2f(x)f(y).$$

Because $n = m = 2$, each of the two voters is drawing two utilities from $\mathcal{D}$, for a total of 4 i.i.d. draws from $\mathcal{D}$. Denote the utilities of voter 1 as $(w, x)$ and the utilities of voter 2 as $(y, z)$ respectively, and consider the case when $x \geq w$ and $y \geq z$, with corresponding ranking $\boldsymbol{\sigma}$. The preference profile is symmetric, and therefore the expected inverse distortions of the two alternatives are equal. We can assume WLOG that alternative 1 has welfare of $y + w$ (ranked second by voter 1 and first by voter 2) and alternative 2 has welfare of $x + z$ (ranked first by voter 1 and second by voter 2). Using the joint pdf for the two sets of two draws, this allows us to calculate the expected inverse distortion of alternative 1 as follows:

$$
\begin{aligned}
\mathbb{E}[\mathrm{invddist}(1, \boldsymbol{\sigma})] &= \int_0^1 \int_0^1 \int_0^x \int_0^y \frac{\max(x+z, y+w)}{y+w} 4f(w)f(z)f(x)f(y) \, dz \, dw \, dy \, dx \\
&\geq 4 \int_0^1 \int_0^1 \int_0^x \int_0^y \frac{x+z}{y+w} f(w)f(z)f(x)f(y) \, dz \, dw \, dy \, dx \\
&= \frac{1}{64} \int_0^1 \int_0^1 \int_0^x \int_0^y \frac{x+z}{y+w} \cdot \frac{1}{(xyzw)^{0.75}} \, dz \, dw \, dy \, dx \\
&\longrightarrow \infty.
\end{aligned}
$$

Therefore no deterministically chosen alternative has finite $\mathbb{E}[\mathrm{invddist}(f, \boldsymbol{\sigma})]$. ∎

## C Randomization and Inverse Distributional Distortion

In Appendix A, we showed that a deterministic rule always achieves the highest expected distortion. For expected inverse distortion as defined in Appendix B, this is no longer the case. In fact, there exist distributions $\mathcal{D}$ supported on $[0, 1]$ such that the best deterministic rule does arbitrarily worse than the best randomized rule.

**Theorem C.1** *For all $\varepsilon > 0$, there exists a distribution $\mathcal{D}$ and preference profile $\boldsymbol{\sigma}$ such that the deterministic rule that maximizes expected inverse distortion has expected inverse distortion less than $\varepsilon$ fraction of that of the best random rule, i.e:*

$$\max_j \mathbb{E}[\mathrm{invddist}(j, \boldsymbol{\sigma})] \leq \varepsilon \cdot \sup_f \mathbb{E}[\mathrm{invddist}(f, \boldsymbol{\sigma})]$$

**Proof.** To prove the desired result, we will show that there exist $n, m, \boldsymbol{\sigma}$, and $\mathcal{D}$ such that there is a random rule that performs at least $1/\varepsilon$ times better than the best deterministic rule. Let $n = m = 100$. Define $\mathcal{D}$ as follows for $d \sim \mathcal{D}$.

$$
d = \begin{cases} \frac{\varepsilon}{2nm} & \text{w.p. } 1 - \frac{1}{nm} \\ 1 & \text{w.p. } \frac{1}{nm} \end{cases}
$$

Consider the uniform preference profile $\boldsymbol{\sigma}$, where every alternative is ranked $i$th by exactly 1 voter for all $i \in [1:n]$. First, note that the rule that chooses an alternative uniformly at random achieves expected inverse distortion at most $m$, as uniformly randomly selecting an alternative will choose the alternative with the maximum social welfare with probability $1/m$. Therefore, the best randomized rule must have expected inverse distortion of at most $m$.

Define $N_j$ to be the total number of voters with utility 1 for alternative $j$. Define $N_{tot} = \sum_{j=1}^{m} N_j$. By construction of $\mathcal{D}$,

$$\Pr(N_{tot} = 0) = \left(1 - \frac{1}{nm}\right)^{nm} \leq 0.4$$

and

$$\Pr(N_j = 0) = \left(1 - \frac{1}{nm}\right)^{n} \geq 0.9$$

Now consider the deterministic voting rule that selects alternative $j$. By a union bound, the probability that $N_{tot} > 0$ and $N_j = 0$ is at least

$$\Pr\left[N_{tot} > 0 \cup N_j = 0\right] \geq 1 - 0.4 - 0.1 = 0.5$$

If $N_{tot} > 0$ and $N_j = 0$, then alternative $j$ has total social welfare of $\frac{\varepsilon}{2m}$ and there is at least one alternative with social welfare greater than 1. Therefore, we can write

$$\mathbb{E}[\mathrm{invddist}(j, \boldsymbol{\sigma})] \geq 0.5 * \frac{1}{\frac{\varepsilon}{2m}} = \frac{m}{\varepsilon}$$

Because the preference profile $\boldsymbol{\sigma}$ is symmetric across voters, the above inequality is true for all $j$. Therefore, the best deterministic rule gives expected inverse distortion that is a factor of $\varepsilon$ worse than that of the rule which selects an alternative uniformly at random. $\blacksquare$

The above result is especially disturbing if we allow ourselves to relax the independent and identically distributed assumption. For example, suppose that among the $nm$ pairs of voters and alternatives, there is a single voter that has value 1 for a single alternative, and every other voter/alternative pair has utility $\frac{\varepsilon}{2nm}$. By symmetry, the best deterministic rule is to choose an arbitrary alternative. This rule has expected inverse distortion at least

$$\frac{m-1}{m}\left(\frac{1}{\frac{\varepsilon}{2m}}\right) \geq \frac{m}{\varepsilon}.$$

In contrast, the randomized rule that chooses one of the $m$ alternatives uniformly at random has expected inverse distortion of at most $m$. Notice that the joint distribution of the social welfare of the chosen alternative and the maximum social welfare across all alternatives is the same for both rules. Therefore, one would expect them to have the same expected inverse distortion value. However, the random rule can have arbitrarily better inverse distortion as shown above. In contrast, these two rules will have the same expected distortion. This is another reason why we chose to define distributional distortion with the ratio in the opposite direction as worst-case distortion.

## D  Proof of Theorem 3.1: Omitted Lemma

**Lemma D.1** *If $x \geq y \geq 0$, then for all $Z_1, Z_2 \geq 0$, the following inequality holds:*

$$\frac{Z_1 + x}{\max(Z_1 + x, Z_2 + y)} \geq \frac{Z_1 + y}{\max(Z_1 + y, Z_2 + x)}.$$

**Proof.** Consider the following two cases.

Case 1: If $Z_1 + x \geq Z_2 + y$, then

$$\frac{Z_1 + x}{\max(Z_1 + x, Z_2 + y)} = 1 \geq \frac{Z_1 + y}{\max(Z_1 + y, Z_2 + x)}$$

Case 2: If $Z_1 + x < Z_2 + y$, then since $x \geq y$ we have that

$$\frac{Z_1 + x}{\max(Z_1 + x, Z_2 + y)} = \frac{Z_1 + x}{Z_2 + y} \geq \frac{Z_1 + y}{Z_2 + x} = \frac{Z_1 + y}{\max(Z_1 + y, Z_2 + x)}.$$

$\blacksquare$

# E   Proof of Theorem 4.1

**Lemma E.1** *For $a \geq 0$ and integer $b \geq 2$ such that $ab \leq 1/5$,*
$$1 - ab \leq (1-a)^b \leq 1 - ab + (ab)^2$$

**Proof.** Expanding the binomial gives:
$$(1-a)^b = 1 - ab + \sum_{k=2}^{b} \binom{b}{k} (-a)^k$$

Examining the last terms, we have that
$$\left| \sum_{k=2}^{b} \binom{b}{k} (-a)^k \right| \leq \frac{a^2 b(b-1)}{2} + \left| \sum_{k=3}^{b} \binom{b}{k} (-a)^k \right|$$
$$\leq \frac{a^2 b^2}{2} + \sum_{k=3}^{b} \binom{b}{k} (-a)^k$$
$$\leq \frac{a^2 b^2}{2} + \sum_{k=3}^{b} (ab)^k$$
$$\leq \frac{a^2 b^2}{2} + \frac{a^3 b^3}{1-ab}$$
$$\leq a^2 b^2.$$

Similarly,
$$\sum_{k=2}^{b} \binom{b}{k} (-a)^k = \frac{a^2 b(b-1)}{2} + \sum_{k=3}^{b} \binom{b}{k} (-a)^k$$
$$= \frac{a^2 b^2}{2} - \frac{a^2 b}{2} + \sum_{k=3}^{b} \binom{b}{k} (-a)^k$$
$$\geq \frac{a^2 b^2}{2} - \frac{a^2 b}{2} - \sum_{k=3}^{b} (ab)^k$$
$$= \frac{a^2 b^2}{2} - \frac{a^3 b^3}{1-ab} - \frac{a^2 b}{2}$$
$$\geq 0.$$

∎

**Proof of Theorem 4.1.** Let $m \geq 5$. Choose any $n > 1$ such that $\frac{\sqrt{n}}{\log(n)} \geq m \geq 2\log(n) + 2$. Define $\mathcal{D}$ as follows: for $u_{ij} \sim \mathcal{D}$,
$$u_{ij} = \begin{cases} n^4 m & \text{w.p. } \frac{1}{n^2 m} \\ n & \text{w.p. } p := e^{-\log(m)/m} - \frac{1}{n^2 m} \\ 1 & \text{w.p. } q := 1 - e^{-\log(m)/m}. \end{cases}$$

Draw $m$ samples from $\mathcal{D}$ and denote them $X_{(1)}, \ldots, X_{(m)}$, in increasing order. The expectation of the largest sample is:
$$\mathbb{E}[X_{(m)}] = \Pr(X_{(m)} = 1) + \Pr(X_{(m)} = n) \cdot n + \Pr(X_{(m)} = n^4 m) \cdot n^4 m$$
$$= q^m + \left( \left(1 - \tfrac{1}{n^2 m}\right)^m - q^m \right) n + \left( 1 - \left(1 - \tfrac{1}{n^2 m}\right)^m \right) n^4 m$$
$$\geq q^m + \left( \left(1 - \tfrac{1}{n^2 m}\right)^m - q^m \right) n + \left( 1 - \left(1 - \tfrac{1}{n^2} + \tfrac{1}{n^4}\right) \right) n^4 m$$
$$= q^m + \left( \left(1 - \tfrac{1}{n^2 m}\right)^m - q^m \right) n + n^2 m - m$$
$$\geq n^2 m - m,$$

where we used Lemma E.1 and the fact that $1 - 1/n^2 m \geq q$.

For every $k < m$, we have that

$$
\begin{aligned}
\mathbb{E}[X_{(k)}] &\leq \mathbb{E}[X_{(m-1)}] \\
&= \Pr(X_{(m-1)} = 1) + \Pr(X_{(m-1)} = n) \cdot n + \Pr(X_{(m-1)} = n^4 m) \cdot n^4 m \\
&\leq n \Pr(X_{(m-1)} \neq n^4 m) + \left(1 - \left(1 - \tfrac{1}{n^2 m}\right)^m - m\left(1 - \tfrac{1}{n^2 m}\right)^{m-1}\left(\tfrac{1}{n^2 m}\right)\right) n^4 m \\
&\leq n + \left(\tfrac{1}{n^2} - \left(1 - \tfrac{m-1}{n^2 m}\right)\left(\tfrac{1}{n^2}\right)\right) n^4 m \\
&\leq n + 2m,
\end{aligned}
$$

where again we used Lemma E.1.

Combining these two inequalities, we have that $\mathbb{E}[X_{(m)}] > n \, \mathbb{E}[X_{(k)}]$ for every $k$. Therefore, one voter ranking a chosen alternative first has higher expected utility than all $n$ voters ranking a chosen alternative second. This implies that if there is a unique alternative that is ranked first by the most voters, then that alternative is the unique expected welfare maximizing alternative.

We consider the following preference profile. Alternative 1 is ranked first by $\frac{n}{m-1} + 1$ voters and ranked $m^{th}$ by $n - \left(\frac{n}{m-1} + 1\right) = n \cdot \frac{m-2}{m-1} - 1$ voters. Alternative 2 is ranked second by all $n$ voters. Finally, all other alternatives are ranked first by at most $\frac{n}{m-1}$ voters.

In this construction, alternative 1 has the most first place ranks and therefore is the unique expected welfare maximizing alternative. We now show that

$$
\mathbb{E}[\mathrm{ddist}(1, \boldsymbol{\sigma})] \leq O\left(\frac{\mathbb{E}[\mathrm{ddist}(2, \boldsymbol{\sigma})]}{m}\right).
$$

Let $\mathcal{E}_1$ be the event that no voter has utility $n^4 m$ for any alternative. By Lemma E.1, we have that

$$
\Pr(\mathcal{E}_1) = \left(1 - \tfrac{1}{n^2 m}\right)^{nm} \geq 1 - \tfrac{1}{n}.
$$

Let $\mathcal{E}_2$ be the event that every voter has utility at least $n$ for alternative 2. Under event $\mathcal{E}_2$, the total welfare of alternative 2 is at least $\mathrm{dsw}(2, \boldsymbol{\sigma}) \geq n^2$. We have that

$$
\begin{aligned}
\Pr(\mathcal{E}_2) &= \Pr(X_{(m-1)} \geq n)^n \\
&\geq \Pr(X_{(m-1)} = n)^n \\
&= \left(1 - q^m - m q^{m-1}(1 - q)\right)^n \\
&\geq \left(1 - q^m - m q^{m-1}\right)^n \\
&\geq \left(1 - \left(\frac{\log(m)}{m}\right)^m - m\left(\frac{\log(m)}{m}\right)^{m-1}\right)^n \\
&\geq \left(1 - \left(\frac{\log(m)}{m}\right)^m - \left(\frac{(m)^{1/(m-1)} \log(m)}{m}\right)^{m-1}\right)^n \\
&\geq \left(1 - \left(\frac{1}{2}\right)^m - \left(\frac{1}{2}\right)^{m-1}\right)^n \\
&\geq \left(1 - \frac{1}{n^2}\right)^n \\
&\geq 1 - \frac{1}{n},
\end{aligned}
$$

where we simplified using that $q \leq \frac{\log(m)}{m}$, $m \geq 5$, and $m \geq 2\log(n) + 2$.

We finally want to establish a tighter upper bound the typical utility of alternative 1 than event $\mathcal{E}_1$ alone provides. Let $N_1$ be the number of voters, among the $n \cdot \frac{m-2}{m-1} - 1$ voters that ranked alternative 1 in last place, who have utility strictly greater than 1 for alternative 1. Then let $\mathcal{E}_3$ be the event that:

$$
N_1 \leq \tfrac{1}{m}\left(n \cdot \tfrac{m-2}{m-1} - 1\right) + \log(n)\sqrt{n}.
$$

We will show $\mathcal{E}_3$ is also likely. To calculate $\Pr(\mathcal{E}_3)$, note that by definition of the probability $p$,

$$\Pr(X_{(1)} > 1) = \left(p + \tfrac{1}{n^2 m}\right)^m = \left(m^{-1/m}\right)^m = \tfrac{1}{m}.$$

Therefore, $N_1 \sim Bin(\tfrac{1}{m}, n \cdot \tfrac{m-2}{m-1} - 1)$. Applying Hoeffding's inequality to this random variable gives:

$$
\begin{aligned}
\Pr(\mathcal{E}_3) &= 1 - \Pr\left(N_1 \geq \tfrac{1}{m}(n \cdot \tfrac{m-2}{m-1} - 1) + \log(n)\sqrt{n}\right) \\
&= 1 - \Pr\left(N_1 - \mathbb{E}[N_1] \geq \log(n)\sqrt{n}\right) \\
&\geq 1 - \exp\left(-2(\log(n)\sqrt{n})^2 / (n \cdot \tfrac{m-2}{m-1} - 1)\right) \\
&\geq 1 - \exp(-2(\log(n)\sqrt{n})^2 / n) \\
&= 1 - \exp(-2\log^2(n)) \\
&\geq 1 - \tfrac{1}{n}.
\end{aligned}
$$

Define $\mathcal{E} := \mathcal{E}_1 \cap \mathcal{E}_2 \cap \mathcal{E}_3$. By inclusion/exclusion and a union bound,

$$
\begin{aligned}
\Pr(\mathcal{E}) &= 1 - \Pr(\neg\mathcal{E}_1 \cup \neg\mathcal{E}_2 \cup \neg\mathcal{E}_3) \\
&\geq 1 - \Pr(\neg\mathcal{E}_1) - \Pr(\neg\mathcal{E}_2) - \Pr(\neg\mathcal{E}_3) \\
&\geq 1 - \frac{3}{n}.
\end{aligned}
$$

We consider the social welfare of alternatives 1 and 2 under event $\mathcal{E}$ to obtain bounds on their respective expected distributional distortions.

Event $\mathcal{E}_3$ implies that the total number of voters that have utility greater than 1 for alternative 1 (including the $\frac{n}{m-1} + 1$ voters who rank alternative 1 first) is at most:

$$\tfrac{1}{m}\left(n \cdot \tfrac{m-2}{m-1} - 1\right) + \log(n)\sqrt{n} + \tfrac{n}{m-1} + 1.$$

Under event $\mathcal{E}_1$, no voter has utility greater than $n$ for any alternative. Therefore, every voter has utility either 1 or $n$ for alternative 1. Since we have an upper bound on the number of voters that have utility $n$ for alternative 1, and trivially at most $n$ voters have utility 1 for alternative 1, we upper bound the total social welfare for alternative 1 by:

$$
\begin{aligned}
\mathrm{dsw}(1, \boldsymbol{\sigma}) &\leq n \cdot \left(\tfrac{1}{m}\left(n \cdot \tfrac{m-2}{m-1} - 1\right) + \log(n)\sqrt{n} + \tfrac{n}{m-1} + 1\right) + 1 \cdot n \\
&\leq \tfrac{n^2}{m} + \log(n)n^{3/2} + \tfrac{2n^2}{m} + n \\
&\leq \tfrac{4n^2}{m} + \log(n)n^{3/2}.
\end{aligned}
$$

Combining this with the fact that $\mathrm{dsw}(2, \boldsymbol{\sigma}) \geq n^2$ under event $\mathcal{E}_2$, we get that the distributional distortion under event $\mathcal{E}$ is at most the following, using that $m \leq \frac{\sqrt{n}}{\log(n)}$:

$$\frac{\tfrac{4n^2}{m} + \log(n)n^{3/2}}{n^2} = \frac{4}{m} + \frac{\log(n)}{\sqrt{n}} \leq \frac{5}{m}.$$

Since $\Pr(\mathcal{E}) \geq 1 - \frac{3}{n}$, the distributional distortion of alternative 1 is upper bounded as follows:

$$\mathbb{E}[\mathrm{ddist}(1, \boldsymbol{\sigma})] \leq \left(1 - \tfrac{3}{n}\right) \cdot \tfrac{5}{m} + \tfrac{3}{n} \cdot 1 \leq \tfrac{8}{m}.$$

Under event $\mathcal{E}$, alternative 2 has total social welfare of $n^2$ and the max social welfare of any alternative is $n^2$. This implies that alternative 2 has distributional distortion 1. The expected distributional distortion of alternative 2 is thus lower bounded as follows:

$$\mathbb{E}[\mathrm{ddist}(2, \boldsymbol{\sigma})] \geq \left(1 - \frac{3}{n}\right) \cdot 1 + \frac{3}{n} \cdot 0 \geq \frac{1}{2}.$$

Therefore,

$$
\begin{aligned}
\mathbb{E}[\mathrm{ddist}(\mathrm{ewmr}, \boldsymbol{\sigma})] &= \mathbb{E}[\mathrm{ddist}(1, \boldsymbol{\sigma})] \\
&\leq \tfrac{16}{m} \mathbb{E}[\mathrm{ddist}(2, \boldsymbol{\sigma})] \\
&\leq \tfrac{16}{m} \mathbb{E}[\mathrm{ddist}(\mathrm{edmr}, \boldsymbol{\sigma})].
\end{aligned}
$$

Note that while this distribution is not supported on $[0, 1]$, simply scaling the values of the distribution by $\frac{1}{n^4 m}$ will make the distribution supported on $[0, 1]$. Furthermore, expected distortion is invariant to scaling the distribution, and therefore the result still holds. ∎

## F   Proof of Theorem 4.2

**Lemma F.1** *Suppose the distribution $\mathcal{D}$ has mean $\mu$ and is supported on $(0, 1]$. Then the expected distributional distortion of the alternative selected by the expected welfare maximizing rule is at least $\mu$. Furthermore, the expected welfare of the expected welfare maximizing alternative is at least $n\mu$.*

**Proof.** First, note that we can add the expected distortion for all $m$ players and apply linearity of expectation to get that

$$\sum_{j=1}^{m} \mathbb{E}[\mathrm{dsw}(j, \boldsymbol{\sigma})] = \sum_{j=1}^{m} \sum_{i=1}^{n} \mathbb{E}[u_{ij}] = nm\mu.$$

Furthermore, the maximum expected social welfare alternative must have expected social welfare at least as large as the average expected social welfare across all alternatives. This implies that there must exist some $j$ such that

$$\mathbb{E}[\mathrm{dsw}(j, \boldsymbol{\sigma})] \geq n\mu.$$

Therefore, the alternative selected by the expected welfare maximizing rule must have expected social welfare of at least $n\mu$. Therefore, using the fact that no alternative can have social welfare of greater than $n$,

$$\begin{aligned}
\mathbb{E}[\mathrm{ddist}(\mathrm{ewmr}, \boldsymbol{\sigma})] &= \mathbb{E}\left[\frac{\mathrm{dsw}(\mathrm{ewmr}, \boldsymbol{\sigma})}{\max_j \mathrm{dsw}(j, \boldsymbol{\sigma})}\right] \\
&\geq \mathbb{E}\left[\frac{\mathrm{dsw}(\mathrm{ewmr}, \boldsymbol{\sigma})}{n}\right] \\
&= \frac{\mathbb{E}[\mathrm{dsw}(\mathrm{ewmr}, \boldsymbol{\sigma})]}{n} \\
&\geq \frac{n\mu}{n} \\
&= \mu.
\end{aligned}$$

∎

The below is a purely technical lemma.

**Lemma F.2** *If $X_1, ..., X_m$ are i.i.d. draws from a continuously differentiable distribution with cdf $F$. Suppose that $\frac{dF(x)}{dx} > 0$ for all $x$. Then for every $s_\varepsilon > 0$ there exists $M$ such that for every $m > M$ and for every $j$,*

$$\mathrm{Var}(X_{(j)}) \leq s_\varepsilon^2.$$

**Proof.** Let $F$ be the cdf of $\mathcal{D}$ and let $F_m$ be the empirical cdf from $m$ samples. Let $|F - F_m| = \sup_x |F(x) - F_m(x)|$. The iterated logarithm law of Smirnov (see Smirnov [1944], Chung [1949] ) states that:

$$\Pr\left(\limsup_{m \to \infty} \frac{\sqrt{m}|F_m - F|}{(2 \log\log(m))^{1/2}} = c \leq \frac{1}{2}\right) = 1.$$

A consequence of this is that for any $\delta > 0$, there exists an $M_1$ such that for all $m \geq M_1$:

$$\Pr\left(|F_m - F| \leq \frac{(2 \log\log(m))^{1/2}}{\sqrt{m}}\right) \geq 1 - \delta. \tag{5}$$

Since $F$ is continuously differentiable it is continuous, and since $\frac{dF}{dx} > 0$ on $(0, 1)$ $F$ is strictly increasing and so its inverse $F^{-1}$ is continuous and well-defined on $[0, 1]$ also. Since $F^{-1}$ is

continuous and bounded on a closed interval, it is uniformly continuous; this means that for all $\varepsilon' > 0$ there is some $\delta' > 0$ such that for all $b \in [0, 1]$, if $|y - b| < \delta'$ then $|F^{-1}(y) - F^{-1}(b)| < \varepsilon'$.

Next, for given $m$, consider the order statistic $X_j$. We note that the $j^{th}$ order statistic is equivalent to $F_m^{-1}(\frac{j}{m})$. For quantiles of the form $b = \frac{j}{m}$, we will suppose that $F_m^{-1}$ is well-defined and $F_m^{-1}(F_m(b)) = b$. For such a point $b$, let $a := F^{-1}(b)$ and let $a_m := F_m^{-1}(b)$.

We now introduce another condition on $m$. Given $s_\varepsilon$, choose $M_2$ such that $\frac{(2 \log \log(M_2))^{1/2}}{\sqrt{M_2}} \leq \delta'$, where $\delta'$ is the threshold given by the uniform continuity of $F^{-1}$ with the choice of $\varepsilon' = s_\varepsilon$. (Since this is a decreasing function of $M_2$, this inequality will hold for all $m \geq M_2$.) By Equation (5) applied to the value $a_m$, we have that

$$|F(a_m) - F_m(a_m)| \leq \delta'$$

with probability at least $1 - \delta$. If this is the case, then by uniform continuity,

$$|F^{-1}(F(a_m)) - F^{-1}(F_m(a_m))| \leq s_\varepsilon$$
$$|F_m^{-1}(b) - F^{-1}(b)| \leq s_\varepsilon,$$

where this last inequality follows by applying the definition of $a_m$.

Using the fact that $X_{(j)}$ is supported on $(0, 1]$, this gives that the variance of $X_{(j)}$ can be bounded as

$$\mathrm{Var}(X_{(j)}) \leq \left( \frac{(2 \log \log(m))^{1/2}}{c\sqrt{m}} \right)^2 + \delta \leq s_\varepsilon^2,$$

where we take $\delta = \frac{s_\varepsilon^2}{2}$ and sufficiently large $m \geq M = \max(M_1, M_2)$. ∎

**Proof of Theorem 4.2.** Recall that $\mathrm{ewmr}(\boldsymbol{\sigma})$ is the expected welfare maximizing alternative. Let $n \geq n_0$, where $n_0$ is the solution to the equation $\frac{\log(n)}{\sqrt{\mu n}} = \varepsilon_0$. By Lemma F.1, the expected welfare maximizing alternative satisfies $\mathbb{E}[\mathrm{dsw}(\mathrm{ewmr}, \boldsymbol{\sigma})] \geq n\mu$. Furthermore, note that the social welfare of any alternative $j$ can be represented as $\mathrm{dsw}(j, \boldsymbol{\sigma}) = \sum_{i=1}^n u_{ij}$ where $u_{ij}$ are $n$ independent random variables that are supported on $(0, 1]$. Therefore, we can apply Hoeffding's theorem to get the following:

$$\begin{aligned}
&\Pr(\mathrm{dsw}(\mathrm{ewmr}, \boldsymbol{\sigma}) \leq (1 - \varepsilon)\mathbb{E}[\mathrm{dsw}(\mathrm{ewmr}, \boldsymbol{\sigma})]) \\
&= \Pr(\mathrm{dsw}(\mathrm{ewmr}, \boldsymbol{\sigma}) - \mathbb{E}[\mathrm{dsw}(\mathrm{ewmr}, \boldsymbol{\sigma})] \leq -\varepsilon \mathbb{E}[\mathrm{dsw}(\mathrm{ewmr}, \boldsymbol{\sigma})]) \\
&\leq \Pr(\mathrm{dsw}(\mathrm{ewmr}, \boldsymbol{\sigma}) - \mathbb{E}[\mathrm{dsw}(\mathrm{ewmr}, \boldsymbol{\sigma})] \leq -\varepsilon n\mu) \\
&\leq e^{-2(\varepsilon n\mu)^2/n} \\
&\leq e^{-2\varepsilon^2 n\mu^2}.
\end{aligned}$$

For any $j \neq \mathrm{ewmr}(\boldsymbol{\sigma})$, we must have $\mathbb{E}[\mathrm{dsw}(j, \boldsymbol{\sigma})] \leq \mathbb{E}[\mathrm{dsw}(\mathrm{ewmr}, \boldsymbol{\sigma})]$. Therefore:

$$\begin{aligned}
&\Pr(\mathrm{dsw}(j, \boldsymbol{\sigma}) \geq (1 + \varepsilon)\mathbb{E}[\mathrm{dsw}(\mathrm{ewmr}, \boldsymbol{\sigma})]) \\
&\leq \Pr(\mathrm{dsw}(j, \boldsymbol{\sigma}) \geq \mathbb{E}[\mathrm{dsw}(j, \boldsymbol{\sigma})] + \varepsilon \mathbb{E}[\mathrm{dsw}(\mathrm{ewmr}, \boldsymbol{\sigma})]) \\
&= \Pr(\mathrm{dsw}(j, \boldsymbol{\sigma}) - \mathbb{E}[\mathrm{dsw}(j, \boldsymbol{\sigma})] \geq \varepsilon \mathbb{E}[\mathrm{dsw}(\mathrm{ewmr}, \boldsymbol{\sigma})]) \\
&\leq \Pr(\mathrm{dsw}(j, \boldsymbol{\sigma}) - \mathbb{E}[\mathrm{dsw}(j, \boldsymbol{\sigma})] \geq \varepsilon n\mu) \\
&\leq e^{-2(\varepsilon n\mu)^2/n} \\
&\leq e^{-2\varepsilon^2 n\mu^2}.
\end{aligned}$$

Let $\mathcal{E}$ be the event that the welfare of the $\mathrm{ewmr}(\sigma)$ alternative is not too small, and that no other alternative has unusually large welfare; $\mathrm{dsw}(\mathrm{ewmr}, \boldsymbol{\sigma}) \geq (1 - \varepsilon)\mathbb{E}[\mathrm{dsw}(\mathrm{ewmr}, \boldsymbol{\sigma})]$ and that for all $j \neq \mathrm{ewmr}(\boldsymbol{\sigma})$, $\mathrm{dsw}(j, \boldsymbol{\sigma}) \leq (1 + \varepsilon)\mathbb{E}[\mathrm{dsw}(\mathrm{ewmr}, \boldsymbol{\sigma})]$. Mathematically,

$$\mathcal{E} = \left\{ \mathrm{dsw}(\mathrm{ewmr}, \boldsymbol{\sigma}) \geq (1-\varepsilon)\mathbb{E}[\mathrm{dsw}(\mathrm{ewmr}, \boldsymbol{\sigma})] \right\} \cap \left\{ \forall j : \mathrm{dsw}(j, \boldsymbol{\sigma}) \leq (1+\varepsilon)\mathbb{E}[\mathrm{dsw}(\mathrm{ewmr}, \boldsymbol{\sigma})] \right\}.$$

Using the two results above, the probability of the event $\mathcal{E}$ can be lower bounded by the following:

$$\Pr(\mathcal{E}) \geq 1 - e^{-2\varepsilon^2 n\mu^2} - e^{-2\varepsilon^2 n\mu^2} = 1 - 2e^{-2\varepsilon^2 n\mu^2}$$

Fix any $j$. Under event $\mathcal{E}$, the distributional distortion of this alternative satisfies the following inequality:

$$\text{ddist}(j, \boldsymbol{\sigma}) = \frac{\text{dsw}(j, \boldsymbol{\sigma})}{\max_j \text{dsw}(j, \boldsymbol{\sigma})}$$

$$\leq \frac{(1 + \varepsilon)\, \mathbb{E}[\text{dsw}(\text{ewmr}, \boldsymbol{\sigma})]}{\max_j \text{dsw}(j, \boldsymbol{\sigma})}$$

$$\leq \frac{1 + \varepsilon}{1 - \varepsilon} \cdot \frac{\text{dsw}(\text{ewmr}, \boldsymbol{\sigma})}{\max_j \text{dsw}(j, \boldsymbol{\sigma})}$$

$$= \frac{1 + \varepsilon}{1 - \varepsilon} \cdot \text{ddist}(\text{ewmr}, \boldsymbol{\sigma})$$

Under event $\neg\mathcal{E}$, the distributional distortion of $j$ is at most 1 by definition. Therefore, we can upper bound the expected distributional distortion of $j$ relative to the expected distributional distortion of $\text{ewmr}(\boldsymbol{\sigma})$ as follows:

$$\mathbb{E}[\text{ddist}(j, \boldsymbol{\sigma})] = \Pr(\mathcal{E}) \cdot \mathbb{E}[\text{ddist}(j, \boldsymbol{\sigma})|\, \mathcal{E}] + \Pr(\neg\mathcal{E}) \cdot \mathbb{E}[\text{ddist}(j, \boldsymbol{\sigma})|\, \neg\mathcal{E}]$$

$$\leq \tfrac{1+\varepsilon}{1-\varepsilon} \cdot \Pr(\mathcal{E}) \cdot \mathbb{E}[\text{ddist}(\text{ewmr}, \boldsymbol{\sigma})|\, \mathcal{E}] + 2e^{-2\varepsilon^2 n \mu^2}$$

$$\leq \tfrac{1+\varepsilon}{1-\varepsilon} \cdot \Big( \Pr(\mathcal{E}) \cdot \mathbb{E}[\text{ddist}(\text{ewmr}, \boldsymbol{\sigma})|\, \mathcal{E}] + \Pr(\neg\mathcal{E}) \cdot \mathbb{E}[\text{ddist}(\text{ewmr}, \boldsymbol{\sigma})|\, \neg\mathcal{E}] \Big) + 2e^{-2\varepsilon^2 n \mu^2}$$

$$\leq \tfrac{1+\varepsilon}{1-\varepsilon} \cdot \mathbb{E}[\text{ddist}(\text{ewmr}, \boldsymbol{\sigma})] + 2e^{-2\varepsilon^2 n \mu^2} \cdot \tfrac{1}{\mu}\, \mathbb{E}[\text{ddist}(\text{ewmr}, \boldsymbol{\sigma})]$$

$$\leq \Big( \tfrac{1+\varepsilon}{1-\varepsilon} + \tfrac{2}{\mu} \cdot e^{-2\varepsilon^2 n \mu} \Big)\, \mathbb{E}[\text{ddist}(\text{ewmr}, \boldsymbol{\sigma})].$$

Take $\varepsilon = \frac{\varepsilon_0}{8} \leq \frac{1}{2}$, which is equal to $\frac{\log(n)}{2\sqrt{\mu n}}$ by our choice of $n$. Then:

$$\tfrac{1+\varepsilon}{1-\varepsilon} + \tfrac{2}{\mu}e^{-2\varepsilon^2 n \mu} \leq 1 + 4\varepsilon + \tfrac{2}{\mu}e^{-2\varepsilon^2 n \mu}$$

$$\leq 1 + \tfrac{\varepsilon_0}{2} + \tfrac{2}{\mu}\sqrt{n}^{-\log(n)}$$

$$\leq 1 + \varepsilon_0.$$

We have shown that for all $j$

$$\mathbb{E}[\text{ddist}(j, \boldsymbol{\sigma})] \leq (1 + \varepsilon_0)\, \mathbb{E}[\text{ddist}(\text{ewmr}, \boldsymbol{\sigma})],$$

and therefore,

$$\mathbb{E}[\text{ddist}(\text{edmr}, \boldsymbol{\sigma})] = \max_j \mathbb{E}[\text{ddist}(j, \boldsymbol{\sigma})] \leq (1 + \varepsilon_0)\, \mathbb{E}[\text{ddist}(\text{ewmr}, \boldsymbol{\sigma})].$$

We have shown that the expected social welfare maximizing rule is an $\varepsilon$-approximation for the expected distributional distortion maximizing rule for sufficiently large $n$. We now show that the same holds for sufficiently large $m$ when $\mathcal{D}$ is continuous. The proof is very similar to the above, except that we use Chebyshev's inequality instead of Hoeffding's inequality.

Note that $\text{dsw}(j, \boldsymbol{\sigma})$ is the sum of $n$ independent random variables, and let $s_\varepsilon^2$ be an upper bound on the variance of any one of these random variables. We can use Lemma F.2 to find a $m_0$ that is suitably large. Specifically, we choose $m_0$ such that $m \geq m_0$ implies $s_\varepsilon^2 \leq \frac{\varepsilon_0^3 \mu^3 n}{256}$.

Because the variance of a sum of independent random variables is the sum of the variances, we can bound $\text{Var}(\text{dsw}(j, \boldsymbol{\sigma})) \leq n s_\varepsilon^2$. Applying Chebyshev's inequality then yields:

$$\Pr(\text{dsw}(\text{ewmr}, \boldsymbol{\sigma}) \leq (1 - \varepsilon)\, \mathbb{E}[\text{dsw}(\text{ewmr}, \boldsymbol{\sigma})])$$

$$= \Pr(\text{dsw}(\text{ewmr}, \boldsymbol{\sigma}) - \mathbb{E}[\text{dsw}(\text{ewmr}, \boldsymbol{\sigma})] \leq -\varepsilon\, \mathbb{E}[\text{dsw}(\text{ewmr}, \boldsymbol{\sigma})])$$

$$\leq \Pr(\text{dsw}(\text{ewmr}, \boldsymbol{\sigma}) - \mathbb{E}[\text{dsw}(\text{ewmr}, \boldsymbol{\sigma})] \leq -\varepsilon n \mu)$$

$$\leq \frac{n s_\varepsilon^2}{(\varepsilon n \mu)^2}$$

$$= \frac{s_\varepsilon^2}{\varepsilon^2 \mu^2 n}.$$

For any $j \neq \mathrm{ewmr}(\boldsymbol{\sigma})$, we must have $\mathbb{E}[\mathrm{dsw}(j, \boldsymbol{\sigma})] \leq \mathbb{E}[\mathrm{dsw}(\mathrm{ewmr}, \boldsymbol{\sigma})]$, and therefore:

$$\begin{aligned}
\Pr(&\mathrm{dsw}(j, \boldsymbol{\sigma}) \geq (1 + \varepsilon) \, \mathbb{E}[\mathrm{dsw}(\mathrm{ewmr}, \boldsymbol{\sigma})]) \\
&\leq \Pr(\mathrm{dsw}(j, \boldsymbol{\sigma}) \geq \mathbb{E}[\mathrm{dsw}(j, \boldsymbol{\sigma})] + \varepsilon \, \mathbb{E}[\mathrm{dsw}(\mathrm{ewmr}, \boldsymbol{\sigma})]) \\
&= \Pr(\mathrm{dsw}(j, \boldsymbol{\sigma}) - E[\mathrm{dsw}(j, \boldsymbol{\sigma})] \geq \varepsilon \, \mathbb{E}[\mathrm{dsw}(\mathrm{ewmr}, \boldsymbol{\sigma})]) \\
&\leq \Pr(\mathrm{dsw}(j, \boldsymbol{\sigma}) - \mathbb{E}[\mathrm{dsw}(j, \boldsymbol{\sigma})] \geq \varepsilon n \mu) \\
&\leq \frac{n s_\varepsilon^2}{(\varepsilon n \mu)^2} \\
&= \frac{s_\varepsilon^2}{\varepsilon^2 \mu^2 n}.
\end{aligned}$$

Using the same logic as in the first part of the proof, this means that we have

$$\mathbb{E}[\mathrm{ddist}(j, \boldsymbol{\sigma})] \leq \left( \frac{1 + \varepsilon}{1 - \varepsilon} + \frac{2 s_\varepsilon^2}{\varepsilon^2 \mu^3 n} \right) \mathbb{E}[\mathrm{ddist}(\mathrm{ewmr}, \boldsymbol{\sigma})]$$

We can upper bound this by taking $\varepsilon = \frac{\varepsilon_0}{8}$ to get:

$$\begin{aligned}
\frac{1 + \varepsilon}{1 - \varepsilon} + \frac{2 s_\varepsilon^2}{\varepsilon^2 \mu^3 n} &\leq 1 + 4\varepsilon + \frac{2 s_\varepsilon^2}{\varepsilon^2 \mu^3 n} \\
&\leq 1 + \frac{\varepsilon_0}{2} + \frac{128 s_\varepsilon^2}{\varepsilon_0^2 \mu^3 n} \\
&\leq 1 + \varepsilon_0.
\end{aligned}$$

As in the first section of the proof, this implies that:

$$\mathbb{E}[\mathrm{ddist}(\mathrm{edmr}, \boldsymbol{\sigma})] = \max_j \mathbb{E}[\mathrm{ddist}(j, \boldsymbol{\sigma})] \leq (1 + \varepsilon_0) \, \mathbb{E}[\mathrm{ddist}(\mathrm{ewmr}, \boldsymbol{\sigma})].$$

∎

## G   Proof of Theorem 5.2

**Lemma G.1** *Let $\mathcal{D} \sim \mathrm{Bern}(p)$ for $0 < \delta < p < 1$. For $m$ i.i.d. draws from $\mathcal{D}$, the expected welfare of order statistics satsify the following inequality:*

- *For $k \leq pm - \delta m$,*
$$\mathbb{E}[X_{(k)}] \leq e^{-2\delta^2 m}.$$

- *For $k \geq pm + \delta m$,*
$$\mathbb{E}[X_{(k)}] \geq 1 - e^{-2\delta^2 m}.$$

**Proof.** Fix $k \leq pm - \delta m$. By Hoeffding's inequality, the probability that $X_{(k)}$ is 1 is bounded by:

$$\begin{aligned}
\Pr(X_{(k)} = 1) &= \Pr(\mathrm{Bin}(m, p) \geq pm + \delta m) \\
&= \Pr(\mathrm{Bin}(m, p) - pm \geq \delta m) \\
&\leq e^{-2\delta^2 m}.
\end{aligned}$$

The only two values that $X_{(k)}$ can take are 0 and 1, so this implies that:

$$\mathbb{E}[X_{(k)}] \leq e^{-2\delta^2 m}.$$

The other direction for $k \geq pm + \delta m$ follows from a symmetric argument. ∎

**Proof of Theorem 5.2**. Fix $0 < \alpha \leq 0.5$. Let $\mu = \frac{\alpha}{4}$ and $\delta = \frac{\alpha^2}{4}$. Further let $\mathcal{D}_1 = \mathrm{Bern}(\mu - \delta)$ and $\mathcal{D}_2 = \mathrm{Bern}(\mu)$. Choose $m$ such that

$$e^{-2m\delta^2} \leq \delta,$$

and let $n = (m-1)$. Define the preference profile $\sigma$ as follows. Let alternative 1 have rank $(\mu - \delta)m$ for all $n$ voters. Let all of the other rankings be evenly divided among the other $m-1$ voters, so that for every $j \in [2 : m]$, alternative $j$ is ranked $k$ by 1 voters for every $k \in [1 : m]$ except for $k = (\mu - \delta)m$.

By Lemma G.1, under distribution $\mathcal{D}_1$, alternative 1 will have expected welfare of:

$$\mathbb{E}[\text{dsw}(1, \boldsymbol{\sigma})] \leq n e^{-2m\delta^2} \tag{6}$$
$$\leq n\delta,$$

by the choice of $m$. By Lemma F.1 there exists at least one alternative $j$ with expected social welfare of at least:

$$\mathbb{E}[\text{dsw}(j, \boldsymbol{\sigma})] \geq n\mu. \tag{7}$$

Similarly for $\mathcal{D}_2$, by Lemma G.1 and the choice of $m$, the social welfare for alternative 1 is

$$\mathbb{E}[\text{dsw}(1, \boldsymbol{\sigma})] \geq (1 - e^{-2m\delta^2})n \tag{8}$$
$$\geq (1 - \delta)n. \tag{9}$$

For $j \neq 1$, let $N_j$ be the set of voters giving $j$ rank $(\mu + \delta)m$ or lower (better). Note that there are a total of $(\mu + \delta)nm$ such rankings across all voters, and $n$ of those positions are occupied by alternative 1. Furthermore, note that the social welfare of any alternative $j$ can be represented as $\text{dsw}(j, \boldsymbol{\sigma}) = \sum_{i=1}^{n} u_{ij}$ where $u_{ij}$ are $n$ i.i.d. random variables that are supported on $[0, 1]$. Therefore, for any $j \neq 1$,

$$|N_j| \leq \frac{(\mu + \delta)nm - n}{m - 1} \leq n(\mu + \delta).$$

Using this, we can then conclude that

$$\mathbb{E}[\text{dsw}(j, \boldsymbol{\sigma})] = \sum_{i=1}^{n} E[u_{ij}]$$
$$= \sum_{i \in N_j} \mathbb{E}[u_{ij}] + \sum_{i \notin N_j} \mathbb{E}[u_{ij}]$$
$$\leq 1 \cdot |N_j| + e^{-2m\delta^2} \cdot (n - |N_j|)$$
$$\leq |N_j| + n e^{-2m\delta^2}$$
$$\leq n(\mu + \delta) + n\delta$$
$$= (\mu + 2\delta)n.$$

Under distribution $\mathcal{D}_1$, by (6) and (7) we have that alternative 1 has expected welfare at most $\frac{\delta}{\mu} = \alpha$ fraction of the expected welfare of the expected welfare maximizing alternative. Under distribution $\mathcal{D}_2$, any alternative $j \neq 1$ has expected welfare at most (using that $\alpha, \delta \leq 1/2$)

$$\frac{\mu + 2\delta}{1 - \delta} \leq 2\mu + 4\delta \leq \alpha$$

fraction of the expected welfare of the expected welfare maximizing alternative. Therefore, we can conclude that under this preference profile, no alternative can be chosen that is an $\alpha$-SWMR for both $\mathcal{D}_1$ and $\mathcal{D}_2$ simultaneously. ∎

**Corollary G.2** *If the distribution can depend on $n, m$, then for every $m$ there exists an $n$, a preference profile, and two distributions $D_1^m$ and $D_2^m$ such that no rule is $\frac{\sqrt{\log(m)}}{m^{1/4}}$-SWMR for both distributions.*

**Proof.** Take $\alpha = \frac{2\sqrt{\log(m)}}{m^{1/4}}$. This is valid as $\delta$ satisfies the necessary $m$ inequality for this choice of $\alpha$. ∎

# H  Proof of Theorem 5.4

Denote the mean and median of $\mathcal{D}$ as $\mu$. Note that for any $k \geq \lceil \frac{m}{2} \rceil$, the expectation of the $k$th order statistic from $m$ i.i.d. draws from a symmetric distribution $\mathcal{D}$ satisfies the lower bound

$$\mathbb{E}[X_{(k)}] \geq \mu. \tag{10}$$

Similarly, for any $k < \lceil \frac{m}{2} \rceil$,

$$\mathbb{E}[X_{(k)}] \leq \mu. \tag{11}$$

Among all voters and alternatives, there are at least $\lceil \frac{mn}{2} \rceil$ pairs of voters and alternatives where the voter ranked the alternative in the top $\lceil \frac{m}{2} \rceil$ of alternatives, regardless of the preference profile $\sigma$. Therefore, by the pigeonhole principle, there must exist at least one alternative that is ranked in the top $\lceil \frac{m}{2} \rceil$ by at least $\frac{1}{m} \lceil \frac{mn}{2} \rceil$ voters.

Define $\text{Top-Half}(\boldsymbol{\sigma})$ as the alternative chosen by the Top-Half approval scoring rule and let let $z$ be the number of voters that rank $\text{Top-Half}(\boldsymbol{\sigma})$ in the top $\lceil \frac{m}{2} \rceil$. By definition $\text{Top-Half}(\boldsymbol{\sigma})$ is the alternative that is ranked in the top $\lceil \frac{m}{2} \rceil$ by the most voters, and so $z \geq \frac{1}{m} \lceil \frac{mn}{2} \rceil$. Using (10), the expected welfare of $\text{Top-Half}(\boldsymbol{\sigma})$ can therefore be lower bounded by

$$\mathbb{E}[\text{dsw}(\text{Top-Half}, \boldsymbol{\sigma})] \geq z\mu.$$

Note that because $\mathcal{D}$ is non-negative and symmetric with median $\mu$, its support must lie within the range $[0, 2\mu]$. By construction, for all $j \neq \text{Top-Half}(\boldsymbol{\sigma})$, alternative $j$ is ranked in the top $\lceil \frac{m}{2} \rceil$ by at most $z$ voters. Furthermore, $\mathbb{E}[X_{(k)}] \leq \mu$ for all $k < \lceil \frac{m}{2} \rceil$. Then for any $j \neq \text{Top-Half}(\boldsymbol{\sigma})$,

$$\mathbb{E}[\text{dsw}(j, \boldsymbol{\sigma})] \leq z \cdot 2\mu + (n - z) \cdot \mu = n\mu + z\mu$$

Combining these two equations with the fact that $z \geq \frac{1}{m} \lceil \frac{mn}{2} \rceil \geq \frac{n}{2}$, we get the following desired result:

$$\frac{\mathbb{E}[\text{dsw}(\text{Top-Half}, \boldsymbol{\sigma})]}{\mathbb{E}[\text{dsw}(\text{ewmr}, \boldsymbol{\sigma})]} \geq \frac{z\mu}{n\mu + z\mu} \geq \frac{1}{3}.$$

∎

# I  Proof of Theorem 5.5

**Proof of Theorem 5.5.** Let $P_1$ be the $\text{Unif}(0, 1)$ distribution and let $P_2$ be the Bernoulli distribution with probability 0.5. Fix $1 > \alpha_0 > 0$. Define $\alpha = \frac{\alpha_0}{4}$. Finally, take $m, n$ such that $\alpha \geq \max \left( \frac{2\log(m)\sqrt{m}+2}{m}, \frac{\log(m)}{\sqrt{m}}, e^{-2\log(m)^2} \right)$. Note that this is simply a technical choice for this proof.

Define the preference profile $\sigma$ as follows, for constants (to be determined later) $p, q$ satisfying $\frac{1}{2} \leq q < p \leq 1$. Suppose alternative 1 is ranked $\frac{m}{2} - \log(m)\sqrt{m}$ by $pn$ voters and ranked last ($m^{th}$) by all other voters. Suppose alternative 2 is ranked first by $qn$ voters and ranked $\frac{m}{2} + \log(m)\sqrt{m}$ by the remaining $(1 - q)n$ voters. For simplicitly, we will assume that $pn, qn, \log(m), \sqrt{m}, \frac{m}{2}$ are integers, and further that $n, pn, qn$ are all divisible by $m - 2$. Suppose all other rankings are distributed as evenly as possible among the remaining voters.

Suppose that the true underlying distribution is $P_1$. The expectation of the $k$th order statistic of a uniform distribution is $\frac{k}{m+1}$. We then get the following lower bounds on the expected social welfare for alternatives 1 and 2:

$$\begin{aligned}
\mathbb{E}[\text{dsw}(1, \boldsymbol{\sigma})] &= pn \cdot \frac{m/2 + \log(m)\sqrt{m}}{m+1} + (1 - p)n \cdot \frac{1}{m+1} \\
&= \frac{pnm/2 + pn\log(m)\sqrt{m} + (1 - p)n}{m+1} \\
&\leq \frac{(p + \alpha)nm}{2(m+1)}, \tag{12}
\end{aligned}$$

$$\mathbb{E}[\mathrm{dsw}(2, \boldsymbol{\sigma})] = qn \cdot \frac{m}{m+1} + (1-q)n \cdot \frac{\frac{m}{2} - \log(m)\sqrt{m}}{m+1}$$

$$= \frac{qnm + (1-q)nm/2 - (1-q)n\log(m)\sqrt{m}}{m+1}$$

$$\geq \frac{(1+q-\alpha)nm}{2(m+1)}. \tag{13}$$

We will now consider what happens when the true underlying distribution is $P_2$. Suppose that this is the case. Then by applying Lemma G.1 with $\delta = \log(m)/\sqrt{m}$ we can bound the expected social welfare for alternative 1 by:

$$(p - \alpha)n \leq pn(1 - e^{-2\log^2(m)})$$
$$\leq \mathbb{E}[\mathrm{dsw}(1, \boldsymbol{\sigma})]$$
$$\leq pn + (1-p)ne^{-2\log^2(m)}$$
$$\leq (p + \alpha)n \tag{14}$$

and similarly for alternative 2 by:

$$(q - \alpha)n \leq qn(1 - e^{-2\log^2(m)})$$
$$\leq \mathbb{E}[\mathrm{dsw}(2, \boldsymbol{\sigma})]$$
$$\leq qn + (1-q)ne^{-2\log^2(m)}$$
$$\leq (q + \alpha)n. \tag{15}$$

For the remaining $m - 2$ alternatives $j \notin \{1, 2\}$, the remaining expected welfare is divided evenly among them. Note that by the divisibility assumptions stated at the beginning, we can exactly evenly divide up the remaining rankings (and therefore expected welfare) among these $m - 2$ alternatives. This allows us to upper bound the expected social welfare for the any alternative $j$ for $j \notin \{1, 2\}$ by:

$$\mathbb{E}[\mathrm{dsw}(j, \boldsymbol{\sigma})] = \frac{1}{m-2}\left(\frac{nm}{2} - E[\mathrm{dsw}(1, \boldsymbol{\sigma})] - E[\mathrm{dsw}(2, \boldsymbol{\sigma})]\right)$$
$$\leq \frac{1}{m-2}\left(\frac{nm}{2} - (p+q)n + 2\alpha n\right)$$
$$\leq \frac{1}{m-2}\left(\frac{nm}{2} - n\right) + \frac{2\alpha n}{m-2}$$
$$\leq \frac{n}{2} + \alpha n$$
$$\leq (q + \alpha)n, \tag{16}$$

since $q \geq 1/2$. Note a symmetric inequality shows that $E[\mathrm{dsw}(j, \boldsymbol{\sigma})] \geq (q - \alpha)n$

If we select alternative 1, then comparing it to alternative 2 under $P_1$ by combining (12) and (13), shows that the expected welfare is at most a $\frac{p+\alpha}{1+q-\alpha}$ fraction of the expected welfare maximizing alternative.

If we select an alternative that is not alternative 1, then by by combining (14), (15), and (16), we see that the expected welfare is at most a $\frac{q+\alpha}{p-\alpha}$ fraction of the expected welfare of the expected-welfare-maximizing alternative.

Therefore, we have a construction such that for every alternative, under some distribution the alternative will be at most a $\frac{q+\alpha}{p-\alpha}$-EWMR or at most a $\frac{p+\alpha}{1+q-\alpha}$-EWMR. From this, we can solve for $p, q$ in the following minimization to make the social welfare approximation for either of these options as low as possible.

$$p, q = \arg\min_{p,q}\left(\max\left(\frac{q+\alpha}{p-\alpha}, \frac{p+\alpha}{1+q-\alpha}\right)\right)$$

Solving this gives that the minimum is achieved at $(p, q) = (\sqrt{0.75}, 0.5)$. For this choice of $p, q$, there is no choice of alternative that achieves at least a

$$\frac{0.5 + \alpha}{\sqrt{0.75} - \alpha} \leq \sqrt{\frac{1}{3}} + 4\alpha \leq \sqrt{\frac{1}{3}} + \alpha_0$$

fraction of the maximum expected welfare alternative if the underlying distribution can be either $P_1$ or $P_2$. Therefore, we have shown that no rule can be better than $(\sqrt{\frac{1}{3}} + \alpha_0)$-EWMR for all symmetric distributions, and taking the limit as $\alpha_0 \to 0$ gives the desired theorem result. ∎

## J   Proof of Theorem 5.7

**Lemma J.1** *If $X_{(m+1-k)}$ is the $(m + 1 - k)^{th}$ order statistic of $m$ draws from a distribution supported on $[0, 1]$ with median $\nu$, then:*

$$\mathbb{E}[X_{(m+1-k)}] \geq 2^{-m} \cdot \nu \cdot \sum_{\ell=k}^{m} \binom{m}{\ell}.$$

**Proof.** Note that a random variable $X \sim \mathcal{D}$ is greater than or equal to the median with probability at least $\frac{1}{2}$. Therefore the probability that the $(m + 1 - k)^{th}$ largest out of $m$ draws is greater than the median is at least:

$$\Pr(X_{(m+1-k)} \geq \nu) \geq \Pr(\text{Bin}(m, \tfrac{1}{2}) \geq k)$$
$$= \sum_{\ell=k}^{m} \binom{m}{\ell} 2^{-m}$$
$$= 2^{-m} \sum_{\ell=k}^{m} \binom{m}{\ell}.$$

Therefore, we can lower bound the expectation of $X_{(m+1-k)}$ by

$$\mathbb{E}[X_{(m+1-k)}] \geq \nu \Pr(X_{(m+1-k)} \geq \nu)$$
$$\geq \nu \cdot 2^{-m} \sum_{\ell=k}^{m} \binom{m}{\ell}.$$

∎

**Proof of Theorem 5.7.** Define $\text{Score}(j)$ to be the score of $j$ under the Binomial Voting:

$$\text{Score}(j) := \sum_{i=1}^{n} \sum_{\ell=r_i}^{m} \binom{m}{\ell}.$$

The scaled sum of the scores of all the alternatives can then be written as

$$\sum_{j=1}^{m} 2^{-m} \text{Score}(j) = n \sum_{k=1}^{m} \sum_{\ell=k}^{m} \binom{m}{\ell} 2^{-m}$$
$$= n \sum_{k=1}^{m} k \binom{m}{k} 2^{-m}$$
$$= n \cdot E \left[ \text{Bin} \left( m, \frac{1}{2} \right) \right]$$
$$= \frac{nm}{2},$$

where the second equality follows by expanding the inner sum. Let $\text{Binom}(\boldsymbol{\sigma})$ be the alternative that is chosen by the Binomial Voting. Then by the pigeonhole principle, the score of $\text{Binom}(\boldsymbol{\sigma})$ must satisfy

$$2^{-m} \cdot \text{Score}(\text{Binom}(\boldsymbol{\sigma})) \geq \frac{n}{2}.$$

We next provide a lower bound on the expected social welfare of an alternative based on the Binomial Voting, and use this to lower bound the expected social welfare of $\mathrm{Binom}(\boldsymbol{\sigma})$. Recall that if an alternative $j$ is ranked $r_1...,r_n$ by the $n$ voters, then Boutilier et al. [2015] tells us that:

$$\mathbb{E}[\mathrm{dsw}(j,\boldsymbol{\sigma})] = \sum_{i=1}^{n} E[X_{(m+1-r_1)}].$$

Applying Lemma J.1, we then find that

$$\mathbb{E}[\mathrm{dsw}(j,\boldsymbol{\sigma})] \geq \sum_{i=1}^{n} 2^{-m}\nu \sum_{\ell=r_i}^{m} \binom{m}{\ell}$$
$$= \nu \cdot 2^{-m} \cdot \mathrm{Score}(j).$$

For $\mathrm{Binom}(\boldsymbol{\sigma})$ this yields

$$\mathbb{E}[\mathrm{dsw}(\mathrm{Binom},\boldsymbol{\sigma})] \geq \nu \cdot 2^{-m} \cdot \mathrm{Score}(\mathrm{Binom}(\boldsymbol{\sigma}))$$
$$\geq \frac{n\nu}{2}$$
$$\geq \frac{\nu}{2} \mathbb{E}[\mathrm{dsw}(\mathrm{ewmr},\boldsymbol{\sigma})],$$

as desired. ∎

## K  Generalized Binomial Voting

**Lemma K.1** *If $X_{(m+1-k)}$ is the $(m+1-k)^{th}$ order statistic of $m$ draws from a distribution $\mathcal{D}$ supported on $[0,1]$, where $Q$ is value of the $p^{th}$ quantile of $\mathcal{D}$, then*

$$\mathbb{E}[X_{(m+1-k)}] \geq Q \cdot \sum_{\ell=k}^{m} \binom{m}{\ell}(1-p)^\ell p^{m-\ell}.$$

**Proof.** As in the proof of Lemma J.1,

$$\Pr(X_{(m+1-k)} \geq Q) \geq \Pr(\mathrm{Bin}(m,1-p) \geq k)$$
$$= \sum_{\ell=k}^{m} \binom{m}{\ell}(1-p)^\ell p^{m-\ell}.$$

Therefore, we can lower bound the expectation of $X_{(m+1-k)}$ by

$$\mathbb{E}[X_{(m+1-k)}] \geq Q \cdot \Pr(X_{(m+1-k)} \geq Q)$$
$$\geq Q \cdot \sum_{\ell=k}^{m} \binom{m}{\ell}(1-p)^\ell p^{m-\ell}.$$

∎

**Proof of Corollary 5.9.** If we define $r_1,...,r_n$ as the rankings of alternative $j$ and

$$\mathrm{Score}(j) := \sum_{i=1}^{n}\sum_{\ell=r_i}^{m} \binom{m}{\ell}(1-p)^\ell p^{m-\ell},$$

then by Lemma K.1,

$$\mathbb{E}[\mathrm{dsw}(j,\boldsymbol{\sigma})] = \sum_{i=1}^{n} \mathbb{E}[X_{(m+1-r_i)}]$$
$$\geq \sum_{i=1}^{n} Q \sum_{\ell=r_i}^{m} \binom{m}{\ell}(1-p)^\ell p^{m-\ell}$$
$$= Q \cdot \mathrm{Score}(j).$$

The sum of the scores of all the alternatives can then be written as:

$$\sum_{j=1}^{m} \text{Score}(j) = n \sum_{k=1}^{m} \sum_{\ell=k}^{m} \binom{m}{\ell} (1-p)^{\ell} p^{m-\ell}$$

$$= n \, \mathbb{E}[\text{Bin}(m, 1-p)]$$

$$= nm(1-p).$$

The rest of the proof follows as in the proof of Theorem 5.7. ∎

**Lemma K.2** *Let $X$ be a random variable supported on $[0,1]$. Choose $T$ values in $[0,1]$, denoted $0 \le Q_1 \le ... \le Q_T \le 1$. Define $Q_0 = 0$ and $Q_{t+1} = 2$. Then*

$$\mathbb{E}[X] \ge \sum_{t=1}^{T} Q_t \cdot \Pr(Q_t \le X < Q_{t+1}).$$

**Proof.** The expectation of $X$ can be written as

$$\mathbb{E}[X] = \int_0^1 \Pr(X \ge x) dx$$

$$= \int_{Q_T}^1 \Pr(X \ge x) dx + \sum_{t=0}^{T-1} \int_{Q_t}^{Q_{t+1}} \Pr(X \ge x) dx$$

$$\ge \sum_{t=0}^{T-1} \int_{Q_t}^{Q_{t+1}} \Pr(X \ge Q_{t+1}) dx$$

$$\ge \sum_{t=0}^{T-1} (Q_{t+1} - Q_t) \cdot \Pr(X \ge Q_{t+1})$$

$$= \sum_{t=1}^{T} Q_t \cdot \Pr(Q_t \le X < Q_{t+1}).$$

∎

**Proof of Theorem 5.11.** Let $X_{(m-k+1)}$ be the $(m-k+1)^{th}$ order statistic from $m$ i.i.d. draws from distribution $\mathcal{D}$. Consider the probability $\Pr(A \le X_{(m-k+1)} < B)$ for $0 \le A \le B \le 1$. Define $p_A := \Pr(X < A)$ and $p_B := \Pr(X < B)$. In order for $A \le X_{(m-k+1)} < B$, there must be at least $k$ draws greater than or equal to $A$, and at most $k-1$ draws greater than or equal to $B$. The probability that there are at least $k$ draws greater than or equal to $A$ is

$$\sum_{\ell=k}^{m} \binom{m}{\ell} (1-p_A)^{\ell} p_A^{m-\ell},$$

and the probability of having at least $k$ draws that are greater than or equal to $B$ is

$$\sum_{\ell=k}^{m} \binom{m}{\ell} (1-p_B)^{\ell} p_B^{m-\ell}.$$

Now using the fact that the second event is a subset of the first event, we have that

$$\Pr(A \le X_{(m-k+1)} < B) = \sum_{\ell=k}^{m} \binom{m}{\ell} (1-p_A)^{\ell} p_A^{m-\ell} - \sum_{\ell=k}^{m} \binom{m}{\ell} (1-p_B)^{\ell} p_B^{m-\ell}. \qquad (17)$$

Therefore the score in Generalized Binomial Voting is a lower bound for the expectation of the $(m-k+1)^{th}$ order statistic, as shown below. We start by applying Lemma K.2 to the order statistic $X_{(m-k+1)}$. By combining this with (17) we get

$$\mathbb{E}[X_{(m-k+1)}] \ge \sum_{t=1}^{T} Q_t \cdot \Pr(Q_t \le X_{(m-k+1)} < Q_{t+1})$$

$$= \sum_{t=1}^{T} Q_t \cdot \left( \sum_{\ell=k}^{m} \binom{m}{\ell} (1-p_t)^{\ell} p_t^{m-\ell} - \sum_{\ell=k}^{m} \binom{m}{\ell} (1-p_{t+1})^{\ell} p_{t+1}^{m-\ell} \right). \qquad (18)$$

Suppose alternative $j$ has rankings $r_1, ..., r_n$ by the $n$ voters, and again define $\mathrm{Score}(j)$ as the total score of this alternative under the Generalized Binomial Voting rule. Then by linearity of expectation and the above inequality,

$$\mathbb{E}[\mathrm{dsw}(j, \boldsymbol{\sigma})] = \sum_{i=1}^{n} \mathbb{E}[X_{(m+1-r_i)}]$$

$$\geq \sum_{i=1}^{n} \sum_{t=1}^{T} Q_t \cdot \left( \sum_{\ell=r_i}^{m} \binom{m}{\ell}(1-p_t)^\ell p_t^{m-\ell} - \sum_{\ell=r_i}^{m} \binom{m}{\ell}(1-p_{t+1})^\ell p_{t+1}^{m-\ell} \right)$$

$$= \mathrm{Score}(j). \tag{19}$$

Now the last step is to find a lower bound for the score of $\mathrm{GBinom}(\boldsymbol{\sigma})$, the index chosen by Generalized Binomial Voting. Note that if we add all of the scores across all of the alternatives, we have that the value can be written as the sum of expectations of the order statistics $Z_{(1)}, ... Z_{(m)}$ of $m$ draws from a Multinomial distribution with $T + 1$ outcomes, with probabilities $\{p_{t+1} - p_t\}_{t=0}^{T}$ and values $0, Q_1, ..., Q_T$ respectively, where we define $p_0 = 0$. This allows us to simplify the sum of scores as follows: In a manner similar to the proof of Theorem 5.7, we have

$$\sum_{j=1}^{n} \mathrm{Score}(j) = n \cdot \sum_{k=1}^{m} \sum_{t=1}^{T} Q_t \left( \sum_{\ell=k}^{m} \binom{m}{\ell}(1-p_t)^\ell p_t^{m-\ell} - \sum_{\ell=k}^{m} \binom{m}{\ell}(1-p_{t+1})^\ell p_{t+1}^{m-\ell} \right)$$

$$= n \cdot \sum_{t=1}^{T} Q_t \left( \sum_{k=1}^{m} \sum_{\ell=k}^{m} \binom{m}{\ell}(1-p_t)^\ell p_t^{m-\ell} - \sum_{k=1}^{m} \sum_{\ell=k}^{m} \binom{m}{\ell}(1-p_{t+1})^\ell p_{t+1}^{m-\ell} \right)$$

$$= n \cdot \sum_{t=1}^{T} Q_t \left( \sum_{k=1}^{m} \mathrm{Pr}[\mathrm{Bin}(m, 1-p_t) \geq k] - \sum_{k=1}^{m} \mathrm{Pr}[\mathrm{Bin}(m, 1-p_{t+1}) \geq k] \right)$$

$$= n \cdot \sum_{t=1}^{T} Q_t \left( \mathbb{E}[\mathrm{Bin}(m, 1-p_t)] - \mathbb{E}[\mathrm{Bin}(m, 1-p_{t+1})] \right)$$

$$= nm \cdot \sum_{t=1}^{T} Q_t (p_{t+1} - p_t). \tag{20}$$

Finally, we conclude as in the proof of Theorem 5.7 that the score of $\mathrm{GBinom}(\boldsymbol{\sigma})$ can be bounded as

$$\mathbb{E}[\mathrm{dsw}(\mathrm{GBinom}, \boldsymbol{\sigma})] \geq \mathrm{Score}(\mathrm{GBinom}(\boldsymbol{\sigma}))$$

$$\geq n \sum_{t=1}^{T} Q_t(p_{t+1} - p_t)$$

$$\geq \sum_{t=1}^{T} Q_t(p_{t+1} - p_t) \, \mathbb{E}[\mathrm{dsw}(\mathrm{ewmr}, \boldsymbol{\sigma})],$$

where we used (19), an averaging argument applied to (20), and that $n \geq \mathbb{E}[\mathrm{dsw}(\mathrm{ewmr}, \boldsymbol{\sigma})]$, since $\mathcal{D}$ takes values in $[0, 1]$. ∎

**Proof of Corollary 5.12**. To prove this result, we need to show that the scores in Generalized Binomial Voting approach the expected values of the order statistics, which are the scores in the ewmr.

Recall in the above proof, the fact that there is an inequality in the equation $\mathbb{E}[\mathrm{dsw}(j, \boldsymbol{\sigma})] \geq \mathrm{Score}(j)$ is because of the inequality in Lemma K.2. That inequality came from the fact that, for random

variable $X$,

$$\mathbb{E}[X] = \int_0^1 \Pr(X \geq x)\, dx$$

$$= \int_{Q_T}^1 \Pr(X \geq x)\, dx + \sum_{t=0}^{T-1} \int_{Q_t}^{Q_{t+1}} \Pr(X \geq x)\, dx$$

$$\geq \sum_{t=0}^{T-1} \int_{Q_t}^{Q_{t+1}} \Pr(X \geq Q_{t+1})\, dx$$

In other words, the score in Generalized Binomial Voting can be thought of as a rectangular lower bound approximation of the integral of expectation of the order statistics of the underlying distribution. By the definition of the Riemann Integral, this rectangle approximation will exactly approach the true integral (making the inequality an equality) as $T \to \infty$. ∎

## L   Sampling

As mentioned in Section 6, we can directly calculate an approximation to the EDMR rule by sampling from the known underlying distribution $\mathcal{D}$. We formalize this result below.

Consider the following sampling algorithm. Take $T$ batches of $nm$ i.i.d. samples each from distribution $\mathcal{D}$. For each batch, assign $m$ samples to each voter $i$, and denote the sorted samples as $X^i_{(1)}, ..., X^i_{(m)}$. Let $r_1^j, ..., r_n^j$ be the rankings of alternative $j$ under $\sigma$. For each $j$, let

$$\hat{\mu}_j = \frac{1}{T} \sum_{t=1}^T \mu_j^t = \frac{1}{T} \sum_{t=1}^T \frac{\sum_{i=1}^n X^i_{(r_i^j)}}{\max_k \sum_{i=1}^n X^i_{(r_i^k)}}$$

be the mean sample distortion of alternative $j$.

**Theorem L.1** *Let $\mathcal{D}$ be bounded on $(0,1]$ with mean $\mu$. For any $\varepsilon, \delta > 0$, take $T = \frac{2\log(\frac{2m}{\delta})}{\mu^2 \varepsilon^2}$. Let $j_{\text{Sampling}} = \arg\max_j \hat{\mu}_j$ be the alternative with the highest mean sample distortion, as defined above. With probability $1 - \delta$, $j_{\text{Sampling}}$ is an $\varepsilon$-approximation of the EDMR:*

$$\mathbb{E}[\text{ddist}(\text{Sampling}, \boldsymbol{\sigma})] \geq (1 - \varepsilon)\, \mathbb{E}[\text{ddist}(\text{edmr}, \boldsymbol{\sigma})].$$

**Proof.** By linearity of expectation, clearly $\mathbb{E}[\hat{\mu}_j] = \mathbb{E}[\mu_j^t] = \mathbb{E}[\text{ddist}(j, \boldsymbol{\sigma})]$. Applying the standard Hoeffding's inequality to $\mu_j$ gives:

$$P\left(|\hat{\mu}_j - \mathbb{E}[\text{ddist}(j, \boldsymbol{\sigma})]| \geq \tfrac{\varepsilon}{2} \cdot \mathbb{E}[\text{ddist}(\text{edmr}, \boldsymbol{\sigma})]\right) \leq 2\exp\left(-T\varepsilon^2\, \mathbb{E}[\text{ddist}(\text{edmr}, \boldsymbol{\sigma})]^2/2\right)$$

$$\leq 2\exp\left(-T\varepsilon^2 \mu^2/2\right)$$

$$= \frac{\delta}{m}.$$

Let $\mathcal{E}$ be the event that for all $j \in [1:m]$,

$$|\hat{\mu}_j - \mathbb{E}[\text{ddist}(j, \boldsymbol{\sigma})]| \leq \tfrac{\varepsilon}{2} \cdot \mathbb{E}[\text{ddist}(\text{edmr}, \boldsymbol{\sigma})].$$

By a union bound, $P(\mathcal{E}) \geq 1 - \delta$. By construction, $j_{\text{Sampling}}$ satisfies $\hat{\mu}_{j_{\text{Sampling}}} \geq \hat{\mu}_{j_{\text{edmr}}}$. Under event $\mathcal{E}$, this implies that

$$\mathbb{E}[\text{ddist}(\text{Sampling}, \boldsymbol{\sigma})] + \tfrac{\varepsilon}{2}\mathbb{E}[\text{ddist}(\text{edmr}, \boldsymbol{\sigma})] \geq \mathbb{E}[\text{ddist}(\text{edmr}, \boldsymbol{\sigma})] - \tfrac{\varepsilon}{2}\mathbb{E}[\text{ddist}(\text{edmr}, \boldsymbol{\sigma})].$$

This equation simplifies to

$$\mathbb{E}[\text{ddist}(\text{Sampling}, \boldsymbol{\sigma})] \geq (1 - \varepsilon)\,\mathbb{E}[\text{ddist}(\text{edmr}, \boldsymbol{\sigma})],$$

as desired.

Note that each of the $T$ batches includes $nm$ samples, so the total number of samples used in this algorithm is $Tnm = \frac{nm\log(\frac{2m}{\delta})}{\mu^2\varepsilon^2}$. ∎

## M   Benchmarks

In this section we elaborate on alternative benchmarks for our results, as briefly mentioned in the discussion. While our results use the EDMR and EWMR as benchmarks, we can also rewrite the statements to instead compare to the actual expected values. We restate Theorem 4.2 and Theorem 5.7 below as Corollary M.1 and Corollary M.2 respectively.

**Corollary M.1** *Assume $\mathcal{D}$ is supported on $[0,1]$ and has constant (relative to $n, m$) mean $\mu$ and variance $s^2$. Then for every $\varepsilon_0 > 0$, there exists $n_0$ such that if $n \geq n_0$, $\mathbb{E}\big[\mathrm{ddist}(\mathrm{ewmr}, \boldsymbol{\sigma})\big] \geq 1 - \varepsilon_0$.*

**Proof.** Define $\mathcal{E}$ as in the proof of Theorem 4.2. Under event $\mathcal{E}$, we have that

$$
\begin{aligned}
\mathrm{ddist}(\mathrm{ewmr}, \boldsymbol{\sigma}) &= \frac{\mathrm{sw}(\mathrm{ewmr}, \boldsymbol{\sigma})}{\max_i \mathrm{sw}(i, \boldsymbol{\sigma})} \\
&\geq \frac{(1-\varepsilon)\,\mathbb{E}[\mathrm{sw}(\mathrm{ewmr}, \boldsymbol{\sigma})]}{(1+\varepsilon)\,\mathbb{E}[\mathrm{sw}(\mathrm{ewmr}, \boldsymbol{\sigma})]} \\
&= \frac{1-\varepsilon}{1+\varepsilon}
\end{aligned}
$$

We can then write

$$
\begin{aligned}
\mathbb{E}[\mathrm{ddist}(\mathrm{ewmr}, \boldsymbol{\sigma})] &\geq \Pr(\mathcal{E}) \cdot \frac{1-\varepsilon}{1+\varepsilon} + \Pr(\neg\mathcal{E}) \cdot 0 \\
&\geq (1 - 2e^{-2\varepsilon^2 n\mu^2}) \frac{1-\varepsilon}{1+\varepsilon} \\
&\geq (1 - 2e^{-2\varepsilon^2 n\mu^2}) \left(1 - \frac{2\varepsilon}{1+\varepsilon}\right) \\
&\geq 1 - 2e^{-2\varepsilon^2 n\mu^2} - 2\varepsilon \\
&\geq 1 - \varepsilon_0
\end{aligned}
$$

where the last inequality comes from taking $n$ suitably large and $\varepsilon = \varepsilon_0/3$. ■

**Corollary M.2** *Let $\mathcal{D}$ be a distribution supported on $[0,1]$ whose largest median is $\nu$, i.e., $\nu = \sup\big\{y \mid \Pr_{x \sim \mathcal{D}}(x \leq y) \leq \frac{1}{2}\big\}$. Then binomial voting achieves expected welfare of at least $\frac{n\nu}{2}$.*

**Proof.** In the last expression of the proof of Theorem 5.7, we exactly have

$$
\begin{aligned}
\mathbb{E}[\mathrm{dsw}(\mathrm{Binom}, \boldsymbol{\sigma})] &\geq \nu \cdot 2^{-m} \cdot \mathrm{Score}(\mathrm{Binom}(\boldsymbol{\sigma})) \\
&\geq \frac{n\nu}{2}
\end{aligned}
$$

■

## N   Total Variation Distance

A desirable result would be that the EWMR for one distribution generalizes well to similar distributions. Unfortunately, this is not the case, and in fact the EWMR for one distribution can have arbitrarily bad expected welfare on an a second distribution that is arbitrarily close in total variation distance.

**Theorem N.1** *For every $0 \leq \delta \leq 1$ and $0 < \varepsilon \leq 1$ there exists $n, m, \sigma$ and two distributions $D_1, D_2$ both supported on $[0,1]$ such that $TV(D_1, D_2) \leq \varepsilon$ and the social welfare maximizing rule on alternative $D_1$ is not a $\delta$-EWMR for $D_2$.*

**Proof.** This follows directly from the construction in the proof of Theorem 5.2 in Appendix G. ■

## N.1   Borda Count

Borda count has been previously shown to be the scoring rule that maximizes expected welfare for the uniform distribution (which we also prove in Lemma N.2. Despite the negative result of Theorem N.1, a somewhat surprising result is that Borda count is also a $(1 - \varepsilon)$-EWMR for distributions close in total variation distance to the uniform distribution. We formalize this result in Theorem N.4.

**Lemma N.2 (Weber [1978])** *Define Borda count as the scoring rule that assigns scores $(1, 2, ..., m)$ to the alternatives in rank order. Then Borda count is equivalent to the expected welfare maximizing rule when the underlying distribution is Uniform[0, 1].*

**Proof.** Let $X_{(1)}, ..., X_{(m)}$ be the order statistics from $m$ i.i.d. draws from the $\mathrm{Unif}(0, 1)$ distribution. Then $\mathbb{E}[X_{(k)}] = \frac{k}{m+1}$. For alternative $j$, let $r_1, ... r_n$ be the ranks of alternative $j$ by voters $1, ..., n$ and let $\mathrm{Score}(j)$ be the Borda count score of alternative $j$. Then by linearity of expectation,

$$
\begin{aligned}
\mathbb{E}[\mathrm{dsw}(j, \boldsymbol{\sigma})] &= \mathbb{E}\left[ \sum_{i=1}^{n} X_{(m+1-r_i)} \right] \\
&= \sum_{i=1}^{n} \mathbb{E}[X_{(m+1-r_i)}] \\
&= \frac{1}{m+1} \sum_{i=1}^{n} m + 1 - r_i \\
&= \frac{1}{m+1} \cdot \mathrm{Score}(j)
\end{aligned}
$$

Therefore, maximizing the expected social welfare is equivalent to maximizing the Borda count score $\mathrm{Score}(j)$. ∎

**Lemma N.3** *Let $U \sim \mathrm{Unif}(0, 1)$ be the uniform distribution and $P$ be a continuous distribution supported on $[0, 1]$ with an invertible cdf such that*

$$
TV(U, P) = \sup_{A \subseteq [0,1]} |P(A) - U(A)| \le \varepsilon.
$$

*Define $X_{(k)}$ as the the $k$th order statistic out of $m$ i.i.d. draws from $P$. Then*

$$
\left| \mathbb{E}[X_{(k)}] - \frac{k}{m+1} \right| \le \varepsilon.
$$

**Proof.** Let $F_P$ and $F_U$ be the cdfs of $P$ and $U$ respectively. Under the assumption, for all $x \in [0, 1]$:

$$
|F_P(x) - F_U(x)| = |F_P(x) - x| \le \varepsilon.
$$

Because the slope of $x$ is 1 and these distributions are supported on $[0, 1]$, this implies that for all $y \in [0, 1]$:

$$
|F_P^{-1}(y) - y| \le \varepsilon.
$$

The probability integral transform for order statistics gives that $X_{(k)} \sim F^{-1}(U_{(k)})$ where $U_{(k)}$ is the $k$th order statistic from $m$ i.i.d. draws from $U$. Using that $\mathbb{E}[U_{(k)}] = \frac{k}{m+1}$, we have the following

result for the expectation of $X_{(k)}$.

$$\left| \mathbb{E}[X_{(k)}] - \frac{k}{m+1} \right| = \left| \mathbb{E}[F^{-1}(U_{(k)})] - \frac{k}{m+1} \right|$$

$$= \left| \int_0^1 F^{-1}(y) f_{U_{(k)}}(y)\, dy - \frac{k}{m+1} \right|$$

$$= \left| \int_0^1 (F^{-1}(y) - y) f_{U_{(k)}}(y)\, dy + \mathbb{E}[U_{(k)}] - \frac{k}{m+1} \right|$$

$$= \left| \int_0^1 (F^{-1}(y) - y) f_{U_{(k)}}(y)\, dy \right|$$

$$\leq \int_0^1 \left| F^{-1}(y) - y \right| f_{U_{(k)}}(y)\, dy$$

$$\leq \int_0^1 \varepsilon f_{U_{(k)}}(y)\, dy$$

$$= \varepsilon.$$

∎

**Theorem N.4** *Let $P$ be a continuous distribution with invertible cdf that is supported on $[0,1]$ such that $TV(U(0,1), P) \leq \varepsilon \leq \frac{1}{10}$. Let* Score *represent the Borda score. Then the Borda count winner is a $5\varepsilon$-approximation of the expected welfare maximizing rule.*

$$\mathbb{E}[\mathrm{dsw}(\mathrm{Borda}, \boldsymbol{\sigma})] \geq (1 - 5\varepsilon)\, \mathbb{E}[\mathrm{dsw}(\mathrm{ewmr}, \boldsymbol{\sigma})].$$

**Proof.** Recall that lemma N.2 shows that

$$\mathrm{Score}(j) = \sum_{i=1}^n m + 1 - r_i^j$$

$$= (m+1)\left( n - \sum_{i=1}^n \frac{r_j^i}{m+1} \right).$$

Define $X_{(m+1-r_i^j)}$ to be the $(m+1-r_i^j)^{th}$ order statistic. Recall that Lemma N.3 states that for all $r_i^j$,

$$\left| \mathbb{E}[X_{(m+1-r_i^j)}] - \frac{m+1-r_i^j}{m+1} \right| \leq \varepsilon.$$

Furthermore,

$$E[\mathrm{dsw}(j, \boldsymbol{\sigma})] = \sum_{i=1}^n \mathbb{E}[X_{(m+1-r_i^j)}].$$

Combining these two equations gives

$$\left| \mathbb{E}[\mathrm{dsw}(j, \boldsymbol{\sigma})] - \sum_{i=1}^n \frac{m+1-r_i^j}{m+1} \right| \leq n\varepsilon,$$

or equivalently,

$$\left| \mathbb{E}[\mathrm{dsw}(j, \boldsymbol{\sigma})] - \frac{\mathrm{Score}(j)}{m+1} \right| \leq n\varepsilon.$$

Because $\mathrm{Borda}(\boldsymbol{\sigma})$ is the alternative that maximizes $\mathrm{Score}(j)$, we can apply this inequality twice to get

$$\mathbb{E}[\mathrm{dsw}(\mathrm{Borda}, \boldsymbol{\sigma})] + n\varepsilon \geq \frac{\mathrm{Score}(\mathrm{Borda}(\boldsymbol{\sigma}))}{m+1}$$

$$\geq \frac{\mathrm{Score}(\mathrm{ewmr}(\boldsymbol{\sigma}))}{m+1}$$

$$\geq \mathbb{E}[\mathrm{dsw}(\mathrm{ewmr}, \boldsymbol{\sigma})] - n\varepsilon.$$

To relate $n\varepsilon$ to $\mathbb{E}[\mathrm{dsw}(\mathrm{ewmr}, \boldsymbol{\sigma})]$, we need to lower bound the expectation. If two distributions supported on $[0, 1]$ have total variation distance of $\varepsilon$, then the means of the two distributions differ by at most $\varepsilon$. Therefore, if $\mu$ is the expected value of $P$, then $\mu \geq 0.5 - \varepsilon$. This means Lemma F.1 implies:

$$\mathbb{E}[\mathrm{dsw}(\mathrm{ewmr}, \boldsymbol{\sigma})] \geq n\mu \geq n\left(0.5 - \varepsilon\right).$$

Therefore, the two equations above combine to:

$$\mathbb{E}[\mathrm{dsw}(\mathrm{Borda}, \boldsymbol{\sigma})] \geq \mathbb{E}[\mathrm{dsw}(\mathrm{ewmr}, \boldsymbol{\sigma})] - 2n\varepsilon$$
$$= \left(1 - \frac{2n\varepsilon}{\mathbb{E}[\mathrm{dsw}(\mathrm{ewmr}, \boldsymbol{\sigma})]}\right)\mathbb{E}[\mathrm{dsw}(\mathrm{ewmr}, \boldsymbol{\sigma})]$$
$$\geq \left(1 - \frac{2n\varepsilon}{\frac{n}{2} - n\varepsilon}\right)\mathbb{E}[\mathrm{dsw}(\mathrm{ewmr}, \boldsymbol{\sigma})]$$
$$\geq (1 - 5\varepsilon)\,\mathbb{E}[\mathrm{dsw}(\mathrm{ewmr}, \boldsymbol{\sigma})]$$

The last inequality comes from taking $\varepsilon \leq \frac{1}{10}$. ∎

# O   Plurality

**Theorem O.1** *For every value of $n, m$ and every distribution $\mathcal{D}$, plurality is an $\alpha$-SWMR for*

$$\alpha = \max\left(\frac{1}{m}, \frac{1}{n}\right)$$

**Proof.** Define $\mathrm{Plurality}(\boldsymbol{\sigma})$ as the plurality winner. Then by definition, $\mathrm{Plurality}(\boldsymbol{\sigma})$ is the top-ranked alternative by at least $\lceil \frac{n}{m} \rceil$ voters. The maximum total expected welfare for an alternative is $n\,\mathbb{E}[X_{(m)}]$, where $\mathbb{E}[X_{(m)}]$ is the expected value of the maximum from $m$ i.i.d. draws from $\mathcal{D}$. Therefore, the expected welfare of $\mathrm{Plurality}(\boldsymbol{\sigma})$ is lower bounded as follows:

$$\mathbb{E}[\mathrm{dsw}(\mathrm{Plurality}, \boldsymbol{\sigma})] \geq \lceil \tfrac{n}{m} \rceil \cdot \mathbb{E}[X_{(m)}]$$

$$\geq \frac{\lceil \frac{n}{m} \rceil}{n} \cdot \mathbb{E}[\mathrm{dsw}(\mathrm{ewmr}, \boldsymbol{\sigma})]$$
$$\geq \max\left(\tfrac{1}{m}, \tfrac{1}{n}\right) \cdot \mathbb{E}[\mathrm{dsw}(\mathrm{ewmr}, \boldsymbol{\sigma})].$$

∎

