# OpenReview forum: "The Distortion of Binomial Voting Defies Expectation"
_NeurIPS.cc/2023/Conference — NeurIPS 2023 poster_

### Official Review · Reviewer_gjAU · 2023-06-30

**Soundness:** 3 good
**Presentation:** 3 good
**Contribution:** 2 fair
**Rating:** 3
**Confidence:** 4

**Summary:**

The authors extend the notion of a distortion of a voting rule to a distributional setting. That is, given a distribution over utilities (that the voters have i.i.d. on the candidates), the authors replace the classic distortion with a random variable that depends on the distribution. The expected value of this random variable is the expected distortion. The authors mostly seek rules that have good expected distortion, as compared to a voting rule with the best one possible.

The two biggest issues with the paper are that the assumed model is highly unrealistic and, worse yet, that it does not seem to lead to any sort of robust conclusions. The assumption that each voter can have a arbitrary (and, possibly, correlated with other voters) preference order, but all the utilities come from the same distribution (i.e., there is a common distribution for each voter and each candidate, except that there is the hidden correlation that utilities that a voter generates for the candidates must respect their preference order) is highly artificial. Then, based on this highly artificial assumption, the authors claim that (under some further, arguably mild assumptions) some specific positional scoring rules (such as m/2-approval and the binomial rule from the title) perform particularly well (however, if I understand correctly, they do not perform _objectively_ well, but simply with respect to the best possible rule, which itself may or may not be very good---I missed if this is clarified in the paper). Now m/2-approval is well understood to be a poor rule because it can be very indecisivle (indeed, if all voters agree that the ranking of the candidates is a > b > c > ... then the rule would not realize that a is the best candidate). The binomial rule is certainly better, but---as argued by the authors---is not too far off m/2-approval.

So, all in all, the authors make a highly questionable assumption and conclude that some rules that do not look too attractive on the outset, optimize some criterion. This is far too weak a conclusion for a paper that expects to be competitive for NeurIPS.

On the positive side, the paper certainly includes high-quality mathematics. Still, I cannot really see it as more than an incremental addition  to the distortion literature (I mean incrementality in terms of conceptual contribution, which---I guess---the authors wanted to stress; on the technical level the paper is far from being incremental).

**Strengths:**

The mathematics behind the results is certainly appealing.

**Weaknesses:**

The underlying assumption of drawing utilities IID is highly unnatural (especially when merged with the assumption that the  utilities still have to respect submitted preference orders, which creates a weird form of correlation).

The conclusions of the paper do not seem to have much practical meaning.

**Questions:**

Q1: You write that "the metric assumption is arguably difficult to justify in most domains of interest". Could you provide the arguments that you have in mind?

Q2: Does your research lead to high-level conclusions beyond "this rule performs well in our setting?". The only thing I could think of is that---perhaps---using some sort of family of distributions that either are IID as yours, or become highly correlated (between the candidates) perhaps one could show that the closer we are to the IID setting, the more natural it is to use positional scoring rules, whereas the more highly correlated are the utilities the better are Condorcet-consistent rules. However, this just a guess and I am not sure how such a result could be obtained. However, generally, resolving the argument between Borda and Condorcet based on the correlations between voters' utilities would certainly be something I would be far more willing to recommend for NeurIPS.

**Limitations:**

The authors admit that their assumption is highly unrealistic, but argue that others make similar assumptions. I understand it is tempting to say so, but given that their work does not seem to give truly valuable conclusions, I think the argument is too weak.

---

> ### Author Rebuttal · Authors · 2023-08-07
>
> **Reviewer comment:**
>
> > The assumption that each voter can have a arbitrary (and, possibly, correlated with other voters) preference order, but all the utilities come from the same distribution (i.e., there is a common distribution for each voter and each candidate, except that there is the hidden correlation that utilities that a voter generates for the candidates must respect their preference order) is highly artificial.
>
> **Response:**
>
> This seems to be your main concern; we believe that it stems from a misunderstanding and that we can provide an effective rebuttal.
>
> When the utilities of the voters for alternatives are drawn i.i.d. from some distribution, the induced preference profile consists of rankings drawn independently and u.a.r. (as you may know, this is called "impartial culture" in social choice). One could define expected distortion with respect to the distribution over utilities, without conditioning on the preference profile. But this is a very "easy" setting for the analysis of distortion, because (especially in the large) all alternatives would have roughly equal social welfare. Therefore, this setting would not differentiate between different voting rules.
>
> When we condition on a preference profile, we are not making an assumption (nor imposing some hidden correlation). Rather, we are setting a tougher requirement. Think of it this way: in most cases, the voting rule would observe a preference profile that is almost symmetric, and then it doesn't really matter which alternative it chooses. But we are requiring the voting rule to do well in *every* situation, regardless of the preference profile it receives as input. This tougher requirement allows us to pinpoint voting rules that truly stand out in terms of their expected distortion (and expected welfare).
>
> To be clear, we are not saying that i.i.d. utilities are not an assumption we'd like to see relaxed in future work — this is something we openly discuss in the paper. But we believe your concern about correlation (which is later referred to as a "weird form of correlation") is unfounded.
>
> **Reviewer comment:**
>
> > Now m/2-approval is well understood to be a poor rule because it can be very indecisivle (indeed, if all voters agree that the ranking of the candidates is a > b > c > ... then the rule would not realize that a is the best candidate).
>
> **Response:**
>
> We do think binomial voting is more attractive, but let us say a few words in defense of m/2-approval. First off, the family of k-approval rules has received quite a bit of attention. A prominent member of the family is the veto rule, which corresponds to (m-1)-approval (each voter vetoes their bottom-ranked alternative). Veto would obviously have the same issue in your example.
>
> From a broader perspective, our model and results provide a novel way of evaluating voting rules. There are many other criteria, including axiomatic desiderata, maximum likelihood estimation under various noise models, distance rationalizability, etc. Your example shows that m/2-approval, without suitable tie-breaking, fails the unanimity axiom. Imposing such additional criteria narrows the search for suitable voting rules; in particular, binomial voting satisfies unanimity. In summary, the new criterion we propose should be seen as another tool in the social choice toolbox, which is meant to be used in conjunction with others.
>
> **Reviewer question:**
>
> > Q1: You write that "the metric assumption is arguably difficult to justify in most domains of interest". Could you provide the arguments that you have in mind?
>
> **Response:**
>
> We note that you are asking us to justify that something is hard to justify :-) But let us respond by way of an anecdote. When one of us talked in 2014 with Elliot Anshelevich, who pioneered the metric view of distortion, the primary example he gave was voting over movies. But we are not convinced that there's a choice of dimensions that gives rise to metric preferences. For example, if the dimensions were quality of the script, quality of acting, etc., then all voters would be located in the same place: the maximum in each dimension. And if one adds dimensions where voters disagree, such as genre, then preferences are no longer single-peaked (imagine a voter who likes comedies and science fiction movies).
>
> Admittedly, this example is a straw man. And don't get us wrong: we love the metric distortion literature and believe it's very valuable. But we also strongly believe in the importance of models that do not impose the metric assumption — especially when one is able to relax the common unit sum assumption, as we do.
>
> **Reviewer question:**
>
> > Q2: Does your research lead to high-level conclusions beyond "this rule performs well in our setting?". The only thing I could think of is that---perhaps---using some sort of family of distributions that either are IID as yours, or become highly correlated (between the candidates) perhaps one could show that the closer we are to the IID setting, the more natural it is to use positional scoring rules, whereas the more highly correlated are the utilities the better are Condorcet-consistent rules.
>
> **Response:**
>
> We believe this question stems from the misunderstanding about correlation, which is addressed above.

---

> > ### Comment · Reviewer_gjAU · 2023-08-18
> > **Response to rebuttal**
> >
> > I certainly see your contribution as something that adds possibly useful results to the distortion literature, but I do not see it as valuable enough to be accepted for NeurIPS. In particular, I am not convinced that your results are useful beyond the distortion literature. If, indeed, you showed a rule that were appealing under a number of criteria _and_ additionally was good with respect to your distortion notion (and, better yet, the distortion view would help you in selecting among several such rules), I would be far more convinced.
> >
> > Q1: "We note that you are asking us to justify that something is hard to justify :-) " <-- No, I am asking you to be responsible for your words. If you write that something "arguably holds" you need to be able to give the arguments.
> >
> > "Weird form of correlation" <-- I see your point of view and I understand that it is better than looking at impartial culture. That said, I think it still involves a form of correlation that is hard to justify. I guess that if you want to convince readers like myself in the future (or in the revised version of the paper) then you would have to make your view as to why this unrealistic assumption makes sense (the argument that "we know it is not really realistic, but is a clear improvement over status quo and this is the best we can do for now" would be good for me, for most venues, but is not sufficient for NeurIPS in my view).
> >
> > All in all,  your papers that two rules are good according to a criterion you invented. Should we use these rules? Should we recommend them? Why is your result important beyond the realm of distortion literature?

---

> > > ### Author Response · Authors · 2023-08-18
> > >
> > > Thank you for your response; we appreciate the opportunity to engage in a discussion with you.
> > >
> > > **Reviewer comment:**
> > >
> > > > If, indeed, you showed a rule that were appealing under a number of criteria and additionally was good with respect to your distortion notion [...] I would be far more convinced.
> > >
> > > **Response:**
> > >
> > > This comment is helpful, and will lead to a stronger presentation of our results. We do believe we can make a convincing case for the general appeal of binomial voting.
> > >
> > > You mentioned the debate between Borda and Condorcet in your original review, so you are likely aware of the arguments in favor of the family of positional scoring rules (which includes Borda). In particular, while no positional scoring rule is Condorcet consistent, they are (when viewed as social choice correspondences) the only voting rules that are anonymous, neutral, and consistent (unifying two profiles with identical winners doesn't change the winner).
> > >
> > > Importantly, binomial voting is not an outlandish voting rule designed purely to achieve low expected distortion. Rather, it is a positional scoring rule, and as such inherits the desirable properties of this family of rules.
> > >
> > > Furthermore, we are aware of very few positional scoring rules that have received attention in their own right, as it's typically hard to justify any specific choice of scores. Examples include plurality, Borda and veto. Another rare example is the harmonic scoring rule of Boutilier et al. [2015]; it was singled out because of its worst-case distortion guarantees, and has become rather well known.
> > >
> > > To summarize, expected distortion helps us pinpoint a new, "special" positional scoring rule, binomial voting, which inherits a number of desirable properties as a member of this family, and additionally guarantees low expected distortion. Much like the harmonic scoring rule, we strongly believe this would be of interest beyond the distortion literature.
> > >
> > > **Reviewer comment:**
> > >
> > > > I think it still involves a form of correlation that is hard to justify. [...] the argument that "we know it is not really realistic, but is a clear improvement over status quo and this is the best we can do for now" [...] is not sufficient for NeurIPS in my view.
> > >
> > > **Response:**
> > >
> > > To clarify, we did not suggest that correlation is "not really realistic" -- our comment regarding an assumption we'd like to relax was about i.i.d. utilities, which is a different matter.
> > >
> > > Instead, our point was that correlation is not even an assumption. Since our previous explanation was unsuccessful, let us offer *two* alternative explanations:
> > >
> > > 1. Consider a policy maker in charge of choosing a voting rule. There are various guarantees such policy maker might wish for the voting rule to satisfy with respect to expected distortion. A first guarantee is that *ex ante*, before an election is held, the distortion is expected to be low. Note that this expectation is over the entire space of voter utility profiles. While this is an appealing guarantee, it is rather weak (and easy to satisfy). Indeed, consider our policy maker not on the day they choose the voting rule, but rather a while later, in a specific election that uses this rule, after the votes have been cast and officially tallied. At this point in time, the ordinal preferences are public knowledge, and, depending on what they are, it might be the case that despite the expected distortion having been low *ex ante*, it turns out that the expected distortion is high *ex post*, that is, conditioned on all of the public information so far — i.e., on the *ordinal* preference profile — the (conditioned) expected distortion is high. A forward-looking policy maker might want a guarantee that this scenario cannot happen, i.e., a worst-case guarantee on the *ex post* expected distortion. This is of course a much stronger guarantee. In particular, it implies the same guarantee on *ex ante* expected distortion, and is considerably harder to satisfy. *This is our guarantee*. To technically phrase a guarantee on the *ex post* expected distortion one needs to condition on the realized publicly announced ordinal preferences, and such conditioning creates correlations.
> > > 2. Without conditioning on preference profiles, expected distortion is defined with respect to the joint distribution over utilities, and the requirement is to do well in expectation over the entire space of utility profiles. Preference profiles partition the space of utility profiles into regions, with each region containing utility profiles that induce the same preference profile. What we're asking is that the voting rule do well in expectation on each and every region of this space, which is a stronger requirement than doing well in expectation over the entire space.
> > >
> > > We feel strongly that our view of this issue of correlation is justified. If the arguments above are still unconvincing to you, we'd be thankful if you would allow us to further elaborate on points that are unclear.

---

> > > > ### Comment · Reviewer_Bk7r · 2023-08-19
> > > > **Distortion as a criteria**
> > > >
> > > > The other review wrote: "If, indeed, you showed a rule that were appealing under a number of criteria and additionally was good with respect to your distortion notion [...] I would be far more convinced."
> > > >
> > > > I am not sure you really responded to this.  The question (for me) is, "Is distortion the primary thing we should be optimizing?"  If so, this is a rather nice paper.  If not, then we are not really optimizing the correct thing.  Simply because you have low distortion does not mean you have a voting rule that should be used.
> > > >
> > > > You rightly claim that you are not inventing a crazy new ridiculous rule.  This is true.  But it still does not mean that we should want to use it.  You say "Rather, it is a positional scoring rule, and as such inherits the desirable properties of this family of rules."  But it also inherits the undesirable properties of positional scoring rules.  In particular, they are easy to manipulate.
> > > >
> > > > I think it is fair to say that in a seven-page report, it is not possible to relitigate distortion as a metric.  Nonetheless, the strength of this paper really does rely on the strength of distortion as a metric.  To me, this is what the other reviewer is asking about.  This is also my hangup for not being a stronger advocate for this paper.

---

> > > > > ### Author Response · Authors · 2023-08-19
> > > > > **Response to Bk7r**
> > > > >
> > > > > It is certainly valid to question whether low (expected) distortion is a compelling argument for using a voting rule. Of course we think it is, and there is a significant literature that builds on the idea that it is, but this is subjective. We do note that our distribution independent results all apply to expected welfare as well, so these results have implications beyond expected distortion, and your opinion need not necessarily rest on the distortion question.
> > > > >
> > > > > However, we disagree that this is the issue at the heart of the point made by Reviewer gjAU. They seem to acknowledge that low distortion is useful, but are not willing to accept a rule with low distortion if it's not "appealing under a number of criteria." You also write that "simply because you have low distortion does not mean you have a voting rule that should be used," and we agree: low distortion is an argument in favor of a rule, but it should be considered holistically alongside other properties.
> > > > >
> > > > > In our latest response to gjAU, we explained why binomial voting is indeed appealing under a number of criteria. Your reaction was that it also has undesirable properties. But, as we know based on the many famous impossibilities in social choice, this is true for any voting rule. For example, if we had shown that Copeland has low expected distortion, a critic could have said that it's susceptible to the no-show paradox (as is any Condorcet consistent rule). Asking for a rule with low expected distortion that is also "appealing under a number of criteria," and has no weaknesses, sets an impossible bar.

---

> > > > > > ### Comment · Reviewer_gjAU · 2023-08-19
> > > > > >
> > > > > > I think that reviewer  Bk7r clarified the issue I have with the paper quite well. Indeed, I am not at all convinced that low distortion is important for a rule---I simply accept the fact that I _might_ be wrong.
> > > > > >
> > > > > > In which real-life settings would you recommend using the binomial rule? A presidential election? An election for a best song? As a scoring rule in Formula 1? Something else?

---

> > > > > > > ### Author Response · Authors · 2023-08-19
> > > > > > > **Response to gjAU**
> > > > > > >
> > > > > > > It was our impression that the issue of correlation played a major role in your criticism, and we responded in great detail to this point. It would be very helpful to know if our response convinced you.
> > > > > > >
> > > > > > > > In which real-life settings would you recommend using the binomial rule? A presidential election? An election for a best song? As a scoring rule in Formula 1? Something else?
> > > > > > >
> > > > > > > A good place to start is group decision making for low-stakes, subjective questions, like where to go for a family vacation, or which movie to go to as a group. In fact, the website Robovote.org, which was online from 2016 to 2022, had two apps, one of which offered free access to voting rules that minimize worst-case distortion precisely for these types of group decisions.

---

> > > > > > > > ### Comment · Reviewer_gjAU · 2023-08-20
> > > > > > > >
> > > > > > > > Thank you for the answers. In the end, I stand by the position that the paper would be a good fit for more specialized venue, but not for NeurIPS. I do not have further questions.

---

### Official Review · Reviewer_M8Ee · 2023-07-07

**Soundness:** 4 excellent
**Presentation:** 4 excellent
**Contribution:** 3 good
**Rating:** 7
**Confidence:** 4

**Summary:**

This paper studies the expected distortion of deterministic voting rules in a distributional setting proposed by Boutilier et al. in 2015. The setting assumes each voter has a random utility for each alternative drawn i.i.d. from a distribution $D$. Then for a given preference profile $\sigma$, the expected welfare of a voting rule is defined as the expectation of winner's total utility conditioned on utilities are consistent with the preference profile $\sigma$. The expected distortion of a voting rule $f$ for a fixed $\sigma$: $ddist(f, \sigma)$ is defined as the expected ratio between total utility of the winner determined by the voting rule and the total utility of the candidate with the highest utility. Note that although utilities are i.i.d. from a distribution, the preference profile can be arbitrary and considered in the worst case.

The first contribution is the authors' demonstration that the majority voting rule results in optimal expected distortion in cases where only two candidates are present. This finding aligns with extensive literature suggesting that majority (or plurality) is the sole "reasonable" voting protocol when faced with a two-candidate scenario.

The paper also explores the connections between expected social welfare and expected distortion. By some standard concentration bounds, the author prove that any voting rule that approximately maximizes expected social welfare also approximately achieves optimal   distortion under mild conditions on the distribution.

Finally, Boutilier et al. [2015] identifies the optimal voting rule in terms of expected welfare. However, this optimal voting rule requires the knowledge of the utility distribution. The main contribution of this paper is that they propose a class of positional voting rules called binomial voting that has good guarantees on expected welfare/distortion for a wide range of distributions. In particular (generalized) binomial voting require no (or limited) information about the distribution.

**Strengths:**

- The distributional model considered in the paper is reasonable. Instead of resorting to questionable random ranking assumptions such as impartial culture, this paper only makes distributional assumptions for utilities but not preferences.

- The new definition expected distortion under this model is well motivated and the authors provide diverse results under various assumptions on the distribution.

- The binomial scoring rule proposed in the paper is novel. It is a nice result that a single voting rule achieves good expected welfare/distortion for a wide range of distributions.

- This paper is very well-written.

**Weaknesses:**

- Though preferences are considered in the worst case, the assumption that all utilities are i.i.d. seems a bit strong.

- The results for $m=2$ and connections between expected welfare and expected distortion are not very surprising and techniques involved are fairly straightforward.

**Questions:**

- Do authors know any results if utilities are still independent but from different distributions for different candidates?

**Limitations:**

The authors addressed all limitations.

---

> ### Author Rebuttal · Authors · 2023-08-07
>
> **Reviewer question:**
>
> > Do authors know any results if utilities are still independent but from different distributions for different candidates?
>
> **Response:**
>
> Relaxing the i.i.d. assumption across either voters $i$ or alteratives $j$ are both interesting directions for future work. We suspect that we may be able to relax the i.i.d. assumption across voters (allowing each voter to have their own distribution for all candidates), as we may be able to leverage similar arguments regarding order statistics. Relaxing the i.i.d. assumption across candidates is likely much more difficult, as the expected utilities of a given voter can no longer be described by order statistics.

---

> > ### Comment · Reviewer_M8Ee · 2023-08-17
> > **Update**
> >
> > Thanks for the response, I have no further questions.

---

### Official Review · Reviewer_3WoR · 2023-07-09

**Soundness:** 3 good
**Presentation:** 3 good
**Contribution:** 3 good
**Rating:** 6
**Confidence:** 5

**Summary:**

This paper studies the distortion of voting rules, which is a measure of how well a voting rule performs with respect to optimal social welfare while having access to limited information about preferences. In particular, the focus is on expected distortion where the underlying utility vectors are drawn from an arbitrary prior distribution and the goal is to design a voting rule that doesn't use any information about such prior distribution, similar in spirit to the design of prior-independent auction mechanisms.

The main approach is to first design an approximate expected welfare maximizing rule (EWMR) and then derive conditions under which EWMR approximates expected distortion maximizing rule (EDMR). When there are only two alternatives ($m=2$), it turns out that the majority is rule is both an EDMR and EWMR. However, this is not true for $m>2$ and the authors consider scoring rule based methods.

The authors show that approving the top half is $1/3$-approximate EWMR for all symmetric distributions. For asymmetric distributions, the authors propose a new scoring called the Binomial voting rule which is $\nu/2$-EDMR for any distribution with median $\nu$. Furthermore, it is also shown that the voting rule can be adapted to incorporate partial knowledge about the underlying distribution e.g. various quantiles.

**Strengths:**

1. I think the notion of prior independent voting rules is quite interesting as the voting rules don't need to depend on the underlying distribution. Furthermore, the results show that prior independent voting rules can achieve constant expected distortion.

2. The connection between EWMR and EDMR voting rules is non-trivial and might turn out to be an interesting approach to designing voting rules that aim to maximize welfare.

3. Finally, the Binomial voting rule is quite interesting and I also liked the result showing that the voting rule can be adapted to incorporate additional information regarding the prior distribution.

**Weaknesses:**

1. The main weakness of the paper is that the utilities of the voters for the $m$ items are assumed to be i.i.d. from the prior distribution. In practice, the utilities of different alternatives are often correlated and this assumption excludes distribution like Gaussian with general covariance matrix. However, the authors have mentioned the limitations in the paper.

2. I am also a bit concerned about the guarantees provided by prior-independent voting rules. Repeated auctions are quite frequent in online platforms and providing bound on performance in expectation makes sense. On the other hand, voting rules are used less frequently -- political elections happen every couple of years, and people's preferences change from one iteration to the next. In such a situation, providing instance dependent bound on distortion is more practical.

**Questions:**

1. If the assumption of symmetric distribution is violated, can you obtain $\alpha$-EWMR for some distribution independent constant $\alpha$? I understand that Theorem 5.2 provides a negative result but the distribution in the theorem needs to depend on $n$, and $m$. Is it still the case for asymmetric distributions independent of $n$, and $m$?

2. Since the distortion of the binomial voting rule depends on the median of the prior distribution, a natural question is why median and why not some other statistic? In particular, if the median of the prior distribution is very small, should the distortion of any voting rule be small as well? Note that, mean can be large in this case as $E[X]$ can be as large as $\textrm{Median}(X) + \sqrt{\textrm{Var}(X)}$.

**Limitations:**

The authors have addressed the limitations of the paper.

---

> ### Author Rebuttal · Authors · 2023-08-07
>
> **Reviewer comment:**
>
> > I am also a bit concerned about the guarantees provided by prior-independent voting rules. Repeated auctions are quite frequent in online platforms and providing bound on performance in expectation makes sense. On the other hand, voting rules are used less frequently -- political elections happen every couple of years, and people's preferences change from one iteration to the next. In such a situation, providing instance dependent bound on distortion is more practical.
>
> **Response:**
>
>
> If we understand correctly, your concern is not about *prior-independent* voting rules, as infrequent elections make it unlikely that much information about the distribution would be available, which strengthens the case for prior independence.
>
> Instead, your concern seems to be about the very idea of optimizing *expected* distortion, and by "instance-dependent bound" you seem to be referring to the worst case over utilities consistent with the given profile. We believe we can provide some useful perspective. First, to say the obvious, this worst-case bound could be extremely bad; in fact, without the unit-sum assumption, there exist preference profiles for which no bound exists. Second, even when given a preference profile where the worst-case bound is reasonable, we would still prefer to optimize expected distortion (assuming these two measures disagree on the outcome), as it is *intuitively* (albeit not formally) likely to lead to an outcome with higher social welfare. (Formalizing this intuition is an interesting problem!)
>
> **Reviewer question:**
>
> > If the assumption of symmetric distribution is violated, can you obtain $\alpha$-EWMR for some distribution independent constant $\alpha$? I understand that Theorem 5.2 provides a negative result but the distribution in the theorem needs to depend on $n$, and $m$. Is it still the case for asymmetric distributions independent of $n$, and $m$?
>
> **Response:**
>
> The distributions in the proof of Theorem 5.2 are both Bernoulli distributions that only depend on $\alpha$. Therefore, if we consider all asymmetric distributions independent of $n$ and $m$, both of these distributions would be included. The proof in the theorem is dependent on $n,m$ because it required $m$ and $n$ to be sufficiently large. Therefore, Theorem 2 could be restated as "For any $\alpha$, there is no rule that is $\alpha$-EWMR for all distributions if we allow $n$ and $m$ to be arbitrarily large, even if we restrict to distributions that are independent of $n,m$". An interesting extension possibly related to your question is, if we restrict to $n,m < 100$ (or any constant), whether $\alpha$-EWMR is possible for some constant $\alpha$. The answer is yes (for example plurality will guarantee $\frac{1}{100}$-EWMR in this case), but characterizing the exact constant is left for future work.
>
>
> **Reviewer question:**
>
> > Since the distortion of the binomial voting rule depends on the median of the prior distribution, a natural question is why median and why not some other statistic? In particular, if the median of the prior distribution is very small, should the distortion of any voting rule be small as well? Note that, mean can be large in this case as $E[X]$ can be as large as $\textrm{Median}(X) + \sqrt{\textrm{Var}(X)}$.
>
> **Response:**
>
> You are correct that the median is not inherently special. The binomial voting rule can be generalized to any quantile or combination of quantiles with the generalized binomial voting rule in Theorem 5.11. If the median of the prior distribution is very small, then using higher quantiles (75%, 95%, etc) would give stronger guarantees with the generalized binomial voting rule. As you point out, medians are not necessarily representative of the mean or higher moments of the distribution. Our results use medians and quantiles because the EWMR calculates the order statistics of the underlying distribution. Moments/means do not provide much information about order statistics, however, quantiles can be used to lower bound order statistics as in Theorems 5.7 and 5.11. This naturally leads to voting rules that use quantiles such as the generalized binomial rule. That being said, it is definitely possible that another rule could use information about moments of the underlying distribution to better approximate the order statistics.

---

> > ### Comment · Reviewer_3WoR · 2023-08-13
> > **Thanks for your response**
> >
> > Dear authors, thank you for your rebuttal and answering my questions. I have read the rebuttal and other reviews as well. Overall, I think this work presents an interesting take on distortion, and would vote to accept the paper.

---

### Official Review · Reviewer_TGPz · 2023-07-14

**Soundness:** 3 good
**Presentation:** 4 excellent
**Contribution:** 4 excellent
**Rating:** 7
**Confidence:** 3

**Summary:**

The paper studies voting rules in the context of expected distortion and expected welfare. In more detail, one of the trending topics in social choice theory is the design of voting rules with low distortion. The underlying assumption for this is that voters have utilities over the alternatives, but only report ordinal preferences. Then, the distortion quantifies the loss of social welfare (=sum of the voters utilities) caused by not knowing the cardinal utilities. To this end, one typically investigates the utility profile (and induced preference profile), where the ratio between the social welfare of the optimal alternative and the alternative chosen by the voting rule is maximal. This is clearly a worst-case measure and allows, without additional assumptions, only for rather negative results. The paper at hand therefore analyzes expected distortion: given a preference profile, the authors assume that, for each voter, the utilities for the alternative are drawn i.i.d from some distribution, and then try to find alternatives that have in expectation a high social welfare (and therefore a good distortion).

Clearly, if we have knowledge about the distribution, it is theoretically possible to compute the alternative that maximizes the expected welfare. The authors thus rely on a model where the prior is not known (and we are thus not in the bayesian setting). In particular, while there is an underlying distribution, it is not known to the mechanism designer. In this setting, the authors then show that for 2 alternatives, the majority rule is the rule that optimizes the expected social welfare and the expected distortion for every underlying distribution. For more than 2 alternatives, the authors furthermore design a new rule called binomial voting (which is a scoring rule relying on the binomial coefficient for the score vector) which always chooses an alternative that has an expected welfare of at least v/2 times the expected welfare of the optimal alternative (where v is the median of the underlying, unknown distribution of utilities). For deriving this result, the authors establish a link between the maximization of expected social welfare and expected distortion. Finally, the authors also discuss several variants of their main result and the limitations of their approach.

**Strengths:**

The paper is well-written and, despite its technical complexity, I found it easy to follow the main ideas of the paper. Furthermore, the paper opens an interesting new direction for going beyond the worst-case distortion in voting by studying the expected distortion. Since the worst-case distortion is known to be prohibitive and the new model seems reasonable, I find the approach quite attractive. Moreover, the results are interesting as they give first, clearly non-trivial bounds for this new setting. Finally, since distortion in voting is a topic that has appeared in NeurIPS before.


**Weaknesses:**

On the negative side, it should be mentioned that many of the main claims of the paper cannot be verified based on the paper itself (as all proofs are deferred to the appendix). (I also did not read the appendix and can therefore not vouch for the correctness of the given results; the proof of Theorem 3.1, which is the only one that is (almost completely) presented in the paper seems correct). Furthermore, I find the comparison to the literature somewhat lacking. For instance, Ebadian et al. (Optimized Distortion and Proportional Fairness in Voting, 2022) suggest several voting rules with close to optimal worst-case distortion and it seems interesting or even necessary to reason why these rules are not good in the given setting. In particular, the harmonic rule by Boutilier et al. (Optimal social choice functions: A utilitarian view, 2015) is also a scoring rule and it thus seems like an interesting question whether it also has a good expected distortion. However, I have to acknowledge that the paper is already quite dense and that the discussion of these results might be too much.



Feedback:

1) I am not a fan of the pun in the title.
2) I think it should be better motivated why the authors use the distribution to draw utilities. This assumption is crucial for the results, not the only way to define expected distortion, and I am not entirely sure about how convincing this model is. In particular, the assumption implies that the preference intensities of voters are (in average) similar, which may not capture reality.
3) Instead of Section 5.2 (which is certainly interesting, but rather dense and technical), one could think of introducing more detailed proof sketches or to discuss the expected distortion of known rules. In principle, I feel that this space is just not used in the most effective way (but the authors may disagree with me here, this is clearly subjective)


**Questions:**

1) Can the authors say something about the expected distortion of known rules? This seems necessary to get some intuition about how strong the results are.
2) Similarly, can the authors say more about upper bounds on expected distortion? Currently, Theorem 5.2 rules out the existence of voting rules with constant expected distortion, but can we, e.g., hope to find a voting rule which is v-EDMR (where v is the median of the distribution) rather than v/2-EDMR?
3) The importance of the results hinges on some implicit assumption. For instance, the main result of the authors is only appealing if the median of the underlying (but unknown) distribution is not too small. Can the authors comment on whether there is some evidence that this might be the case?

---

> ### Author Rebuttal · Authors · 2023-08-07
>
> **Reviewer question:**
>
> > Can the authors say something about the expected distortion of known rules? This seems necessary to get some intuition about how strong the results are.
>
> **Response:**
>
> Since expected distortion is defined with respect to a specific distribution and preference profile, we would need to know these parameters as well as the rule of interest. Given a specific distribution, preference profile, and voting rule, we can directly calculate or approximate the expected distortion using nested integrals or Monte Carlo sampling.
>
> That being said, our results do connect back to some known rules. For uniform distributions (or distributions close to uniform), Borda Count is the expected welfare maximizing rule and therefore a good approximation of the expected distortion maximizing rule. For the two alternatives case, Theorem 3.1 also shows that plurality does maximize distortion. For more than two alternatives, Theorem N.1 shows that plurality is an $\alpha$-EDMR rule for $\alpha = \max(\frac{1}{n}, \frac{1}{m})$.
>
> One reasonable (and interesting!) question for future work could be, what is the *worst* distribution and preference profile for a specific rule, and how does the expected distortion compare to that of the EDMR in this setting? Our guess would be that for most rules, there is some distribution/preference profile pair for which the rule performs very poorly relative to the EDMR. If so, this would imply that our decision of whether or not to use a certain voting rule (e.g. harmonic) should depend on our hypothesis of the underlying distribution. This makes a lot of sense -- we might want to use Borda Count when the underlying distribution is approximately uniform, but not when the underlying distribution is approximately Bernoulli. We could also choose to use a voting rule with distribution independent guarantees such as binomial voting.
>
>
> **Reviewer question:**
>
> > Similarly, can the authors say more about upper bounds on expected distortion? Currently, Theorem 5.2 rules out the existence of voting rules with constant expected distortion, but can we, e.g., hope to find a voting rule which is v-EDMR (where v is the median of the distribution) rather than v/2-EDMR?
>
> **Response:**
>
> We found proving upper bounds (such as no rule can achieve $\nu$-EDMR) to be significantly harder than proving lower bounds. While we do not have a tight upper bound for the best $\alpha$-EDMR rule, the example in the proof of Theorem 5.5 can give some insight into a possible upper bound. In that proof, the two distributions (which happen to be both symmetric) have medians of $0.5$ and $1$ respectively. The proof shows that it is impossible to achieve a $\sqrt{1/3}$-EDMR for both of these distributions. Therefore, no rule can do better than a $1.14\nu$-EDMR where $\nu$ is the median. Clearly, this still does not rule out the possiblity of a $\nu$-EDMR rule, but a more tailored example than the above would almost certaintly give stronger results.
>
>
> **Reviewer question:**
>
> >The importance of the results hinges on some implicit assumption. For instance, the main result of the authors is only appealing if the median of the underlying (but unknown) distribution is not too small. Can the authors comment on whether there is some evidence that this might be the case?
>
> **Response:**
>
> While the binomial voting rule does rely on the median being not too small, the generalized binomial voting rule can give stronger results that depend on higher quantiles. If the median is too low in a specific application for the binomial rule to give a strong result, then using higher quantiles in the generalized binomial voting rule will improve the guarantees. Therefore, one solution is to use a high enough quantile in the generalized binomial rule to get a useful guarantee. If even the higher quantiles are too small to give interesting guarantees, then the distribution has mass concentrated at $0$. A distribution with mass mostly concentrated at $0$ does require a substantially different approach, but this case may be "easier" because most voters have utility $0$ for most alternatives.

---

> > ### Comment · Reviewer_TGPz · 2023-08-18
> > **Thank you!**
> >
> > We would like to thank the authors for replying to our questions. We have no further comments or questions at this point.

---

### Official Review · Reviewer_J4Js · 2023-07-27

**Soundness:** 3 good
**Presentation:** 4 excellent
**Contribution:** 3 good
**Rating:** 5
**Confidence:** 3

**Summary:**

This work introduces the notion of the expected distortion of voting rules. The setup involves n voters and m alternatives. Each voter $i$ has a utility $u_{ij}$ for alternative j, which induces a ranking $\sigma_i$ of the alternatives, where $j$ is ranked higher than $k$ in $\sigma_i$ only if $u_{ij} \geq u_{ik}$. A deterministic voting rule $f$ only has access to the ordinal preferences profiles $\boldsymbol{\sigma}$ instead of the actual underlying utilities $\mathbf{u}$. In the standard worst-case analysis, the distortion of a voting rule is defined as the social welfare incurred by the information loss, i.e.,$$\sup_{\boldsymbol{\sigma}} \sup_{\mathbf{u}: \mathbf{u} \triangleright \boldsymbol{\sigma}} \frac{\max_{j \in A} sw(j,\mathbf{u})}{sw(f(\boldsymbol{\sigma},\mathbf{u}))}.$$
This work considers a different setting. Given a preference profile $\boldsymbol{\sigma}$, a utility vector consistent with the preference profile $\mathbf{u}$ is drawn from some distribution $D$ as follows: Each voter $i$ draws $m$ utilities i.i.d from $D$, and these $m$ utilities are assigned from highest to lowest to the $m$ alternatives according to the order of $\sigma_i$. The paper then defines the distributional distortion of a voting rule $f$ for a given preference $\boldsymbol{\sigma}$ as a random variable $$ddis(f,\boldsymbol{\sigma}) = \frac{sw(f(\boldsymbol{\sigma}), \mathbf{u})}{\max_{j \in A} sw(j,\mathbf{u})}.$$ Subsequently, the expected distortion of $f$ for a given $\boldsymbol{\sigma}$ is the expectation of this ratio, taken over the random process described above.
Similarly, it defines distributional social welfare $dsw(f, \sigma) = sw(f(\boldsymbol{\sigma}), \mathbf{u})$ and expected social welfare. Unlike the worst-case distortion, the benchmarks the paper considers is agnostic to the actual underlying utility profile, but rather the best voting rules that can know only to the distribution $D$ the utilities are drawn from. The best such rules are termed EDMR (expected-distortion-maximizing-rule) and EWMR (expected-welfare-maximizing-rule) for the expected distortion and expected social welfare objective, respectively.

Regarding the results, for the two-alternative case, the paper shows that the Majority rule is both EDMR and EWMR. For multiple agents, by restricting attention to distributions that are independent of $n$ and $m$, and for sufficiently large $n$ or sufficiently large $m$, the expected distortion of an EWMR is a $1-\epsilon$ approximation of the expected distribution of an EDMR. The paper then attempts to approximate the expected welfare which in terms to approximate the expected distortion of EDMR, under the stated conditions. To achieve this, they first present a negative result, demonstrating that it is impossible for a single voting rule to achieve any constant $\alpha$ approximation of the expected social welfare for all distributions supported on $[0,1]$. To overcome this negative result, the paper shows that for symmetric distributions, a scoring rule that assigns a score of 1 to alternatives ranking in the highest half position and 0 to others, and outputs the alternative with the highest score, achieves an expected welfare 1/3 of the welfare of EWMR. IT also show that no voting rule can achieve a $\sqrt{1/3}$ approximation for this setting. Another way to overcome the aforementioned negative result is by assigning a score of $\sum_{\ell = k}^{m} {m\choose\ell}$ to an alternative ranked $k$. The paper demonstrates that this smoother version of the top-half score rule is a $v/2$ approximation to the EWMR, where $v$ is the largest median.

**Strengths:**

The paper is well organized. The high-level idea of moving beyond worst-case analysis for distortion is novel and of great interest. The observation of the connection between the optimal expected welfare and optimal expected distortion under different distributional assumptions is interesting. The distributional-independent adaptation of the scoring rule of Boutilier et al [2015] is intuitive (in a good way) and clever.

**Weaknesses:**

I find the benchmark to be relatively weak, and the positive results are somewhat marginal. Additionally, there are some correctness issues in the proofs.

To expand on the first point regarding the benchmarks, let us recall that the standard worst-case distortion measures the greatest difference between the social welfare of a voting rule that only has access to the ordinal preference and the optimal social welfare of the utilities that are consistent with the preference. This work relaxes the measure in two ways: First, the ratio is now measured in expectation, which aligns well with the idea of the paper. Second, the benchmarks are voting rules that are also agnostic to the underlying utilities but can have knowledge of the distributions from which the values are drawn (EDMR and EWMR). However, it is not clear to me why the second relaxation is needed or interesting, and no motivation is provided regarding the choice of such benchmarks. In fact, without the second relaxation, we can define the expected distortion of a voting rule for $D$ as follows:
$$\inf_{\boldsymbol{\sigma}} \mathbb{E}\left[\frac{sw(f(\boldsymbol{\sigma},\mathbf{u}))}{\max_{j \in A} sw(j,\mathbf{u})}\right].$$
Results regarding the above evaluation seem to better capture the ''average-case analysis'' the paper refers to. To this end, it is unclear how well the EDMR (possibly distribution-dependent) performs with respect to the above benchmarks, which is also interesting and arguably a more natural question to tackle first before distribution-independent voting rules.

Regarding the second point, please note that the positive results in section 5 are all with respect to the expected social welfare considered by Boutilier et al. [2015]. The results presented in this work are stronger since they are distribution-independent, but they are weaker since the distributions are assumed to be bounded (supported on [0,1]). As far as I know, no such restriction is needed for Boutilier et al.'s results to hold. To translate these expected social welfare results into expected distortion results, one needs to further restrict distributions to the ones that are independent with respect to both $n$ and $m$ and with sufficiently large $n$ or $m$, which weakens the positive results. Additionally, none of the bounds in section 5 are tight.

There are a few incorrect arguments in the proofs, the most important one I spotted being in the proof of Lemma F1. The first sentence states that by linearity of expectation, $\mathbb{E}[dsw(j, \sigma)] = \sum^n_{i = 1} \mathbb{E}[u_{ij}] = n\mu$ for all $j$, where $\mu$ is the mean of distribution $D$. The second equality seems to be incorrect. Consider a simple example with 2 alternatives and 2 agents with $D$ being Uniform$[0,1]$, and alternative $1$ is ranked second by any $i$. Then $u_{i1}$ will be the first-order statistic out of two draws for both agents, making $\mathbb{E}[u_{ij}] = 1/3$ instead of $=\mu = 1/2$. This, in turn, makes the proof of F1 invalid. With that said, I believe the statement of Lemma F1 is correct. A possible alternative proof is as follows: since $\sum_{j} dsw(j,\sigma) = \sum_{j}\sum_{i} u_{ij} = mn\mu$, and the maximum $dsw$ is always weakly higher than the average $dsw = n\mu$. Similarly, in the proof of theorem 4.2, the claim that the social welfare of any alternative $j$ can be represented as $dsw(j,\sigma) = \sum^n_{i=1} u_{ij}$, where $u_{ij}$ are $n$ i.i.d. random variables, is also incorrect since $j$ could be ranked in different positions for different voters $i$, and $u_{ij}$s are therefore only independent, not identical. This issue is again not major, as Hoeffding’s inequality only requires the variables to be independent and bounded. However, these false claims hinder my confidence in the rest of the proofs that I did not check to the same level of detail.

**Questions:**

Is there any results or intuition regarding how bad is the worst expected distortion for EDMR for different $D$? For example,for the special case of two alternatives, what's the worst expected distortion for majority for different $D$?

It seems like all the utilities i.i.d drawn from the same distribution is need for a lost of the concentration bounds to go through. Does the results still hold if voters are drawing the values for candidates i.i.d from $D_i$?

Regrading the assumption that D is supported on [0,1], it is true that for the results to go through one only need the utilities to be bounded? Or the maximum of the distribution indeed affects the guarantees?

**Limitations:**

The assumptions are distributions for the positive results to hold are rather restricted. The results are not tight.

---

> ### Author Rebuttal · Authors · 2023-08-07
>
> **Reviewer comment:**
>
> > I find the benchmark to be relatively weak. [...] We can define the expected distortion of a voting rule for $D$ as follows: $$\inf_{\boldsymbol{\sigma}} \mathbb{E}\left[\frac{sw(f(\boldsymbol{\sigma},\mathbf{u}))}{\max_{j \in A} sw(j,\mathbf{u})}\right].$$ Results regarding the above evaluation seem to better capture the ''average-case analysis'' the paper refers to. To this end, it is unclear how well the EDMR (possibly distribution-dependent) performs with respect to the above benchmarks, which is also interesting and arguably a more natural question to tackle first before distribution-independent voting rules.
>
> **Response:**
>
> To paraphrase, you're asking why we aren't providing direct expected distortion (or expected welfare) bounds. This benchmark was indeed our starting point, but in the course of 9 months of working on this project (from September 2022 until May 2023) we shifted focus to approximating the EDMR and EWMR, where we could paint a much more complete picture.
>
> Importantly, our proofs can be adapted in order to restate our results in terms of direct bounds on expected distortion and welfare. For example, the following statements follow from our proofs:
>
> *Thm 4.2 restated:* For every $\epsilon > 0$, there exists $n_0$ such that if $n \ge n_0$, then the expected distortion of the EWMR is at least (1- $\epsilon$).
>
> *Thm 5.7 restated:* Let $\mathcal{D}$ be a distribution supported on $[0,1]$ whose largest median is $\nu$. Then binomial voting achieves expected welfare of at least $\nu n/2$.
>
> Note that both statements are true for all $\sigma$, and therefore also hold when expected distortion is defined with an $\inf$ as in your comment.
>
> In light of your comment, we absolutely agree that more discussion of the benchmarks is needed. In our revision, we commit to adding 1-2 paragraphs about this to the paper, as well as including an appendix to elaborate on the technical connection between the benchmarks. We believe that such a revision would address your concern and would be well within the scope of a conference revision.
>
> **Reviewer comment:**
>
> > There are a few incorrect arguments in the proofs, the most important one I spotted being in the proof of Lemma F1. The first sentence states that by linearity of expectation, $\mathbb{E}[dsw(j, \sigma)] = \sum^n_{i = 1} \mathbb{E}[u_{ij}] = n\mu$ for all $j$, where $\mu$ is the mean of distribution $D$. The second equality seems to be incorrect. [...] Similarly, in the proof of theorem 4.2, the claim that the social welfare of any alternative $j$ can be represented as $dsw(j,\sigma) = \sum^n_{i=1} u_{ij}$, where $u_{ij}$ are $n$ i.i.d. random variables, is also incorrect since $j$ could be ranked in different positions for different voters $i$, and $u_{ij}$s are therefore only independent, not identical.
>
> **Response:**
>
> Both issues are valid — thanks for catching them and kudos on reading Appendix F! The latter issue is a typo ("i.i.d." should be "independent"), but the former issue is an embarrassing, albeit easily fixable, mistake. The correct argument is actually commented out in our LaTex file; it appears that in the last proofreading round, one of us introduced the mistake by attempting to slightly simplify the argument. The paper (including appendices) had been carefully proofread by multiple authors over several days before submission. We therefore hope that the minor mistake will be seen as a fluke that doesn't reflect on the soundness of our results.
>
> **Reviewer question:**
>
> > Is there any results or intuition regarding how bad is the worst expected distortion for EDMR for different $D$? For example,for the special case of two alternatives, what's the worst expected distortion for majority for different $D$?
>
> **Response:**
>
> This question is partially addressed above. We do not know the answer your specific question about two alternatives, but not for lack of trying — we spent a while looking at regular and MHR distributions in this context.
>
> **Reviewer question:**
>
> > It seems like all the utilities i.i.d drawn from the same distribution is needed for a lot of the concentration bounds to go through. Do the results still hold if voters are drawing the values for candidates i.i.d from $D_i$?
>
> **Response:**
>
> Relaxing the i.i.d. assumption across either voters $i$ or alteratives $j$ are both interesting directions for future work. We suspect that we may be able to relax the i.i.d. assumption across voters (allowing each voter to have their own distribution for all candidates), as we may be able to leverage similar arguments regarding order statistics. Relaxing the i.i.d. assumption across candidates is likely much more difficult, as the expected utilities of a given voter can no longer be described by order statistics.
>
> **Reviewer question:**
>
> > Regarding the assumption that D is supported on [0,1], it is true that for the results to go through one only need the utilities to be bounded? Or the maximum of the distribution indeed affects the guarantees?
>
> **Response:**
>
> Bounded utilities are sufficient for the results to go through — we chose to focus on $[0,1]$ to make the proofs clearer mathematically, that is, the assumption is made purely for ease of exposition. The maximum of the distribution does not affect our guarantees, as expected distortion is invariant to scaling of the distribution.

---

> > ### Comment · Reviewer_J4Js · 2023-08-17
> >
> > Thank you for the response. I have no further questions at this point.

---

### Official Review · Reviewer_Bk7r · 2023-07-27

**Soundness:** 4 excellent
**Presentation:** 4 excellent
**Contribution:** 3 good
**Rating:** 5
**Confidence:** 3

**Summary:**

This paper studies the concept of expected distortion for voting rules, where distortion measures the worst-case ratio between the maximum social welfare and the rule's welfare. Expected distortion considers the expectation over consistent utility profiles drawn from an underlying i.i.d. distribution. The paper shows majority is optimal for two alternatives. For more alternatives, expected welfare maximization approximates expected distortion maximization for large electorates or alternatives. A novel voting rule called binomial voting is proposed and shown to approximate expected welfare maximization in a distribution-independent manner. Its guarantee depends on the median of the distribution.

**Strengths:**

Provides an interesting perspective by analyzing distortion in an average-case Bayesian setting rather than worst-case.
Establishes asymptotic equivalence between expected distortion and welfare maximization.
Binomial voting is intuitive and has strong guarantees dependent on the median. Approaches expected welfare maximization.
Solid theoretical analysis with proofs of key hardness and approximation results.  Does a good job exploring the space.


**Weaknesses:**

Assumption of i.i.d. utilities may be unrealistic for some domains like politics. Correlated utilities more reflective of reality.
The paper oversells itself.  In the abstract it works for “all distributions”; in the intro, i.i.d. distributions; and then in the main body consistent utility profiles drawn where each value is drawn from some  underlying i.i.d. distribution.
I think I am in the minority on this, but I don’t love distortion as something to optimize.  It assumes truthful voting, which is very dubious in positional voting schemes (like is suggested in the paper).


**Questions:**

I got lost in the proof of Theorem 3.1.  In particular, could not see why a coupling argument was necessary (or sufficient).  I am not suggesting that the proof is wrong, rather (likely) unclear in its current condensed form (it could just be a me thing, but I am not a newbie to complex proofs).

Could you help me out here?

---

> ### Author Rebuttal · Authors · 2023-08-07
>
> **Reviewer comment:**
>
> > The paper oversells itself. In the abstract it works for “all distributions”; in the intro, i.i.d. distributions; and then in the main body consistent utility profiles drawn where each value is drawn from some underlying i.i.d. distribution.
>
> **Response:**
>
> Thank you for letting us know this was unclear, and we are happy to augment the abstract with further details! The different wordings here are because we provide more specific details about the same model as the paper progresses, and not an attempt to oversell the paper.
>
> To clarify, the use of the term "all distributions" in the intro refers to the fact that the underlying distribution D (from which i.i.d samples are drawn) can be any distribution. In addition, consistent utility profiles are necessary when conditioning on a preference profile, which we believe is a harder and more interesting setting than taking an expectation over all preference profiles. This is because without conditioning on a preference profile, all alternatives have roughly the same social welfare (especially in the large), which obviates the need for a good voting rule. If interested, we provide further discussion of this point in our response to reviewer gjAU.
>
>
>
> **Reviewer comment:**
>
> > I got lost in the proof of Theorem 3.1. In particular, could not see why a coupling argument was necessary (or sufficient). I am not suggesting that the proof is wrong, rather (likely) unclear in its current condensed form (it could just be a me thing, but I am not a newbie to complex proofs). Could you help me out here?
>
> **Response:**
>
> Here is a higher level overview of how we used a coupling argument in Theorem 3.1. Please let us know if there are any parts that are still unclear!
>
> In Theorem 3.1, we want to show that the majority winner also has weakly higher expected distortion than the  majority loser. To do this, it is sufficient to show that the expected distortion of alternative 1 when it is ranked first $k$ times is weakly lower than the expected distortion of alternative 1 when it is ranked first $k+1$ times. This is exactly the equation on page 5 following the sentence beginning with "Recall that...". Showing this is sufficient because, by definition, the majority winner is ranked first at least as many times as the majority loser.
>
> This brings us to the coupling argument itself. Define event $A$ as the event that alternative 1 is ranked first $k$ times, and define event $B$ as the event that alternative 1 is ranked first $k+1$ times. The only difference between these two events is that one voter prefers alternative 2 under event $A$ and prefers alternative 1 under event $B$. Therefore, because these two events only differ by a single voter, we can use a coupling argument to compare the expected distortion of alternative 1 under these two events. Assume WLOG that voter $n$ is the differing voter. Fixing the values of the utilities of the first $n-1$ voters (which are the same under both event $A$ and $B$), the only voter that changes the expected distortion is the $n$th voter. This allows us to use the law of total expectation to separate out the randomness of this $n$th voter from the randomness of the rest of the voters. Therefore, the coupling argument is that these first $n-1$ voters are coupled between the two events, with the only change coming from the $n$th voter. The proof concludes with the algebraic equations on the top of page 6, which show exactly the desired inequality between the expectation of the distortion under event $A$ versus event $B$.

---

> > ### Comment · Reviewer_Bk7r · 2023-08-19
> > **Thanks**
> >
> > Thank you for the thoughtful rebuttal.

---

### Decision · Program_Chairs · 2023-09-21

**Decision:**

Accept (poster)

**Comment:**

This paper considers the distortion of voting schemes. Several reviewers have pointed out that distortion, without further justification, is not a very compelling metric --- in particular, the paper would greatly benefit from some important real-world examples where the proposed voting rule and metric would be of importance. The authors are encouraged to add a thorough discussion of applicability.

Still, many reviewers agree that the mathematical results are non-trivial and interesting --- in fact, a majority of the reviewers has a positive view of the paper.

In my assessment, the paper merits acceptance.